# Terahertz Néel spin-orbit torques drive nonlinear magnon dynamics in antiferromagnetic Mn$_2$Au

Y. Behovits [1,2] ✉, A. L. Chekhov [1,2], S. Yu. Bodnar [3,4], O. Gueckstock [1,2], S. Reimers [3], Y. Lytvynenko [3,5], Y. Skourski [6], M. Wolf [2], T. S. Seifert [1,2], O. Gomonay [3], M. Kläui [3], M. Jourdan [3] & T. Kampfrath [1,2] ✉

Antiferromagnets have large potential for ultrafast coherent switching of magnetic order with minimum heat dissipation. In materials such as Mn$_2$Au and CuMnAs, electric rather than magnetic fields may control antiferromagnetic order by Néel spin-orbit torques (NSOTs). However, these torques have not yet been observed on ultrafast time scales. Here, we excite Mn$_2$Au thin films with phase-locked single-cycle terahertz electromagnetic pulses and monitor the spin response with femtosecond magneto-optic probes. We observe signals whose symmetry, dynamics, terahertz-field scaling and dependence on sample structure are fully consistent with a uniform in-plane antiferromagnetic magnon driven by field-like terahertz NSOTs with a torkance of $(150 \pm 50)$ cm$^2$ A$^{-1}$ s$^{-1}$. At incident terahertz electric fields above 500 kV cm$^{-1}$, we find pronounced nonlinear dynamics with massive Néel-vector deflections by as much as 30°. Our data are in excellent agreement with a micromagnetic model. It indicates that fully coherent Néel-vector switching by 90° within 1 ps is within close reach.

Antiferromagnets offer great potential for robust, ultrafast and space- and energy-efficient spintronic functionalities[1–6]. Remarkably, in a recently discovered class of metallic antiferromagnets with locally broken inversion symmetry, coherent rotation of the Néel vector **L** should be possible by a simple application of electrical currents. This fascinating phenomenon is driven by the Néel spin-orbit torques (NSOTs)[7–9] that arise from staggered spin-orbit fields at the antiferromagnetically coupled spin sublattices[10,11]. So far, switching studies of CuMnAs and Mn$_2$Au showed indications of NSOTs[12–15] but also a strong and possibly dominant heat-driven reorientation of **L**[16,17].

To reveal direct signatures of NSOTs and gauge their potential for ultrafast coherent antiferromagnetic switching, electric fields at terahertz frequencies are particularly interesting because they are often resonant with long-wavelength antiferromagnetic magnons[18–23]. Consequently, terahertz NSOTs should require significantly smaller current amplitudes to modify antiferromagnetic order and, consequently, mitigate unwanted effects such as Joule heating. Studying terahertz NSOTs should also provide fundamental insights into key parameters such as torkance and the frequency and lifetime of long-wavelength magnons in the novel antiferromagnets.

Probing antiferromagnetic responses is highly nontrivial because, unlike the magnetization of ferromagnets, the Néel vector **L** cannot be controlled by external magnetic fields below several tesla in most compounds[24–26]. Consequently, experimental separation of magnetic and nonmagnetic dynamical effects is challenging[27,28]. Moreover, the

[1]Department of Physics, Freie Universität Berlin, 14195 Berlin, Germany. [2]Department of Physical Chemistry, Fritz-Haber-Institut der Max-Planck-Gesellschaft, 14195 Berlin, Germany. [3]Institute of Physics, Johannes-Gutenberg-Universität Mainz, 55099 Mainz, Germany. [4]Physikalisch-Chemisches Institut, Ruprecht-Karls-Universität Heidelberg, 69120 Heidelberg, Germany. [5]Institute of Magnetism of the NAS and MES of Ukraine, 03142 Kyiv, Ukraine. [6]Hochfeld-Magnetlabor Dresden (HLD-EMFL), Helmholtz-Zentrum Dresden-Rossendorf, 01328 Dresden, Germany. ✉e-mail: yannic.behovits@fu-berlin.de; tobias.kampfrath@fu-berlin.de

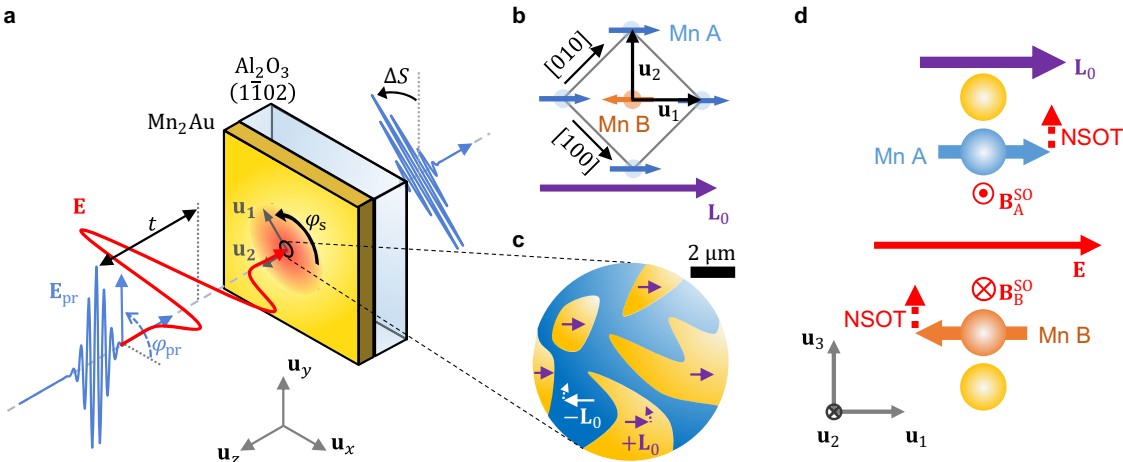

**Fig. 1 | Schematic of experiment and samples. a** A linearly polarized phase-locked terahertz pump pulse (red) with electric field $\mathbf{E} = E\mathbf{u}_x$ is normally incident onto an antiferromagnetic $Mn_2Au$ thin film. The resulting spin dynamics are monitored by an optical probe pulse (blue) with field $\mathbf{E}_{pr}$ by measuring its polarization change $\Delta S$ (rotation and ellipticity) behind the sample vs. delay time $t$. The probe-polarization angle $\varphi_{pr} = \sphericalangle(\mathbf{E}_{pr}, \mathbf{u}_x)$, sample azimuth $\varphi_s = \sphericalangle(\mathbf{u}_1, \mathbf{u}_x)$ and terahertz field polarity ($\pm\mathbf{E}$) can be varied in the laboratory frame ($\mathbf{u}_x, \mathbf{u}_y, \mathbf{u}_z$). The sample-fixed frame is given by $\mathbf{u}_1 = [110]/\sqrt{2}$, $\mathbf{u}_2 = [\bar{1}10]/\sqrt{2}$ and $\mathbf{u}_3 = \mathbf{u}_z = [001]$. **b** Schematic of the $Mn_2Au$ unit cell seen along $\mathbf{u}_z = \mathbf{u}_3$. The local magnetic moments on the Mn A and Mn B sites lead to the sublattice magnetizations $\mathbf{M}_A$ and $\mathbf{M}_B$, respectively. The Néel vector $\mathbf{L} = \mathbf{M}_A - \mathbf{M}_B$ (purple arrow) equals $\mathbf{L}_0$ before pump excitation. For simplicity, the Au atoms are omitted. **c** In the magnetically prealigned sample, the in-plane distribution $\mathbf{L}_0(x,y)$ consists of regions with $\mathbf{L}_0 \uparrow\uparrow +\mathbf{u}_1$ and $\mathbf{L}_0 \uparrow\uparrow -\mathbf{u}_1$, resulting in a pattern of 180° domains. Dashed arrows indicate a possible pump-induced deflection $\Delta\mathbf{L}$. **d** An in-plane electric field $\mathbf{E}$ induces staggered spin-orbit fields $\mathbf{B}_A^{SO} = -\mathbf{B}_B^{SO}$. For $\mathbf{E} \| \mathbf{L}_0$, the resulting NSOTs on Mn A and Mn B moments are maximum and directed out-of-plane ($\| \mathbf{u}_3$).

small size of antiferromagnetic domains[24,29] implies that the spatial average of the Néel vector and the NSOTs vanishes.

In this work, we report on terahertz-pump magneto-optic-probe experiments on $Mn_2Au$ thin films (Fig. 1a). We find birefringence signals linear in the incident transient terahertz electric field. They can consistently be assigned to a uniform, strongly damped and coherent antiferromagnetic magnon at 0.6 THz that is excited by field-like NSOTs. When the terahertz field inside the $Mn_2Au$ film exceeds 30 kV cm$^{-1}$, the Néel-vector dynamics become significantly nonlinear. Comparison to an analytical model allows us to extract values of NSOT torkance, magnon frequency and Gilbert damping, all of which are consistent with previous predictions and experiments. We deduce that the Néel vector $\mathbf{L}$ is transiently deflected by as much as 30° at the maximum peak field of 40 kV cm$^{-1}$.

Our results imply that terahertz electric fields and NSOTs can drive coherent nonlinear magnon dynamics in $Mn_2Au$, thereby bringing coherent switching, a central goal of antiferromagnetic spintronics[1–6], in close reach. Indeed, extrapolation of our data indicates that coherent rotation of $\mathbf{L}$ by 90° can be achieved by terahertz pulses with a moderately increased peak field strength of around 120 kV cm$^{-1}$ inside $Mn_2Au$.

## Results and discussion
### Experiment
As schematically shown in Fig. 1a, we excite $Mn_2Au$ thin films with intense phase-locked single-cycle terahertz pulses to drive spin dynamics and magneto-optically monitor them with a femtosecond probe pulse.

$Mn_2Au$ is a metallic collinear antiferromagnet with a high Néel temperature above 1000 K. It has 2 spin sublattices A and B with magnetization $\mathbf{M}_A$ and $\mathbf{M}_B$, resulting in the net magnetization $\mathbf{M} = \mathbf{M}_A + \mathbf{M}_B$ and Néel vector $\mathbf{L} = \mathbf{M}_A - \mathbf{M}_B$ (Fig. 1b). In equilibrium, one has $\mathbf{M} = \mathbf{M}_0 = 0$ and $\mathbf{L} = \mathbf{L}_0$. $Mn_2Au$ is an easy-plane antiferromagnet with strong out-of-plane ($\mathbf{u}_3 = [001]$) and a 1–2 orders of magnitude smaller biaxial in-plane anisotropy field[24,30–32]. Consequently, the

vectors $\mathbf{M}_A$ and $\mathbf{M}_B$ align along one of the easy axes $\mathbf{u}_1 = [110]/\sqrt{2}$ or $\mathbf{u}_2 = [\bar{1}10]/\sqrt{2}$ (Fig. 1b, c). Due to locally broken inversion symmetry and strong spin-orbit coupling, an electric current induces staggered spin-orbit fields at the A and B sites, which exert NSOTs on the localized magnetic moments[11] (Fig. 1d).

We study 2 epitaxially grown $Mn_2Au$(001) thin films with an $Al_2O_3$ cap layer (thickness of 3 nm) on $Al_2O_3(1\bar{1}02)$ substrates (500 μm). We choose a $Mn_2Au$ thickness of 50 nm, which is optimum with regard to sample crystallinity (favoring thicker films) and terahertz and optical beam transmission (favoring thinner films). In the as-grown samples, the 4 equilibrium directions of the Néel vector $\mathbf{L}_0$ are at 0°, 90°, 180°, and 270° relative to $\mathbf{u}_1$[24,29]. The resulting domains have a size of the order of 1 μm with approximately equal distribution over the film plane and a small strain-induced preference along one easy axis[33].

Following growth, one sample is subject to an intense magnetic-field pulse[24,34,35] that aligns most domains at 0° or 180° (Fig. 1c). For test purposes, we also consider $Mn_2Au$ | Py samples in which the volume-averaged Néel vector $\langle\mathbf{L}_0\rangle$ is oriented parallel to the magnetization of the exchange-coupled ferromagnetic Py (permalloy $Ni_{80}Fe_{20}$) layer[29].

In our ultrafast setup (Fig. 1a and Supplementary Notes 1 and 2), the terahertz pump and optical probe beam are both normally incident onto the sample. The driving terahertz field $F = (\mathbf{E}, \mathbf{B})$ contains the electric component $\mathbf{E} = E\mathbf{u}_x$, whose waveform $E(t)$ is shown in Fig. 2a, and the magnetic component $\mathbf{B} = B\mathbf{u}_y$, which exhibits the same shape as $E(t)$. In free space, $E$ reaches peak values up to 600 kV cm$^{-1}$, which is, however, reduced to 6% inside the $Mn_2Au$ film. The resulting spin dynamics are monitored by linearly polarized optical probe pulses[36] with the polarization plane at an angle $\varphi_{pr}$ relative to the laboratory axis $\mathbf{u}_x$ (Fig. 1a). The detected signals $\Delta S(t, F, \mathbf{L}_0)$ are the pump-induced probe-polarization rotation and ellipticity vs. pump-probe delay $t$.

The sign of the pump field ($\pm F$) and the local Néel vector ($\pm\mathbf{L}_0$) can be reversed approximately by, respectively, 2 wire-grid polarizers and rotation of the sample azimuth $\varphi_s = \sphericalangle(\mathbf{u}_1, \mathbf{u}_x)$ by 180°. In the following, we use the labels $\pm F$ and $\pm\mathbf{L}_0$ as a shorthand for the respective experimental configurations.

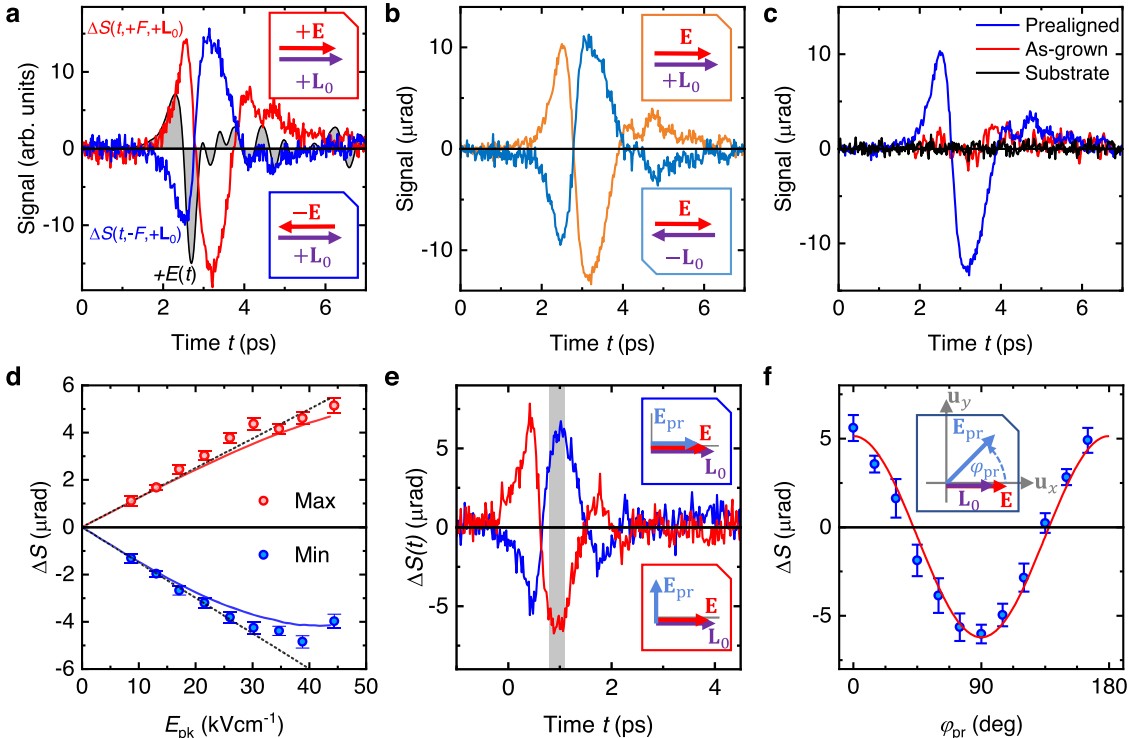

**Fig. 2 | Signatures of terahertz spin dynamics. a** Pump-induced probe polarization rotation $\Delta S(t,+F,\mathbf{L_0})$ (red) and $\Delta S(t,-F,\mathbf{L_0})$ (blue) vs. pump-probe delay $t$ measured for opposite polarities of the terahertz pump field $F = (\mathbf{E},\mathbf{B})$ in the prealigned sample with $\varphi_s = \varphi_{pr} = 0°$. The gray-shaded area shows the terahertz electric field $E(t)$ for reference. The incident peak terahertz field in air is 250 kV cm⁻¹, corresponding to $E_{pk} = 15$ kV cm⁻¹ inside the sample. **b** Signal components $\Delta S(t,+\mathbf{L_0})$ ($\varphi_s = 0°$, orange solid line) and $\Delta S(t,-\mathbf{L_0})$ ($\varphi_s = 180°$, blue) odd in the driving terahertz field [Eq. (1)]. **c** Signal component $\Delta S(t) = [\Delta S(t,+\mathbf{L_0}) - \Delta S(t,-\mathbf{L_0})]/2$ odd in driving field $F$ and Néel vector $\mathbf{L_0}$ for prealigned (blue line) and as-grown (red) sample. The black line shows the signal $\Delta S$ odd in $F$ from the bare $Al_2O_3$ substrate for $\varphi_s = 0°$. **d** Maximum (red dots) and minimum value (blue dots) of signal $\Delta S(t)$ odd in $F$ and $\mathbf{L_0}$ vs. peak terahertz electric field $E_{pk}$ inside the sample. Data was taken from a different sample region than in panels (**a**)–(**c**), yielding smaller signal magnitudes. The underlying signals are displayed in Supplementary Note 6. The red and blue solid line are a model calculation [Eqs. (4) and (17)]. The dotted black lines indicate the linear approximation of the model. **e** Odd-in-$F$ signal $\Delta S(t,+\mathbf{L_0})$ for $E_{pk} = 15$ kV cm⁻¹ and $\varphi_s = \varphi_{pr} = 0°$ (blue) and 90° (red). The insets indicate the probe polarization in the lab frame. For these measurements, a second $Al_2O_3$ substrate compensates the birefringence of the first substrate. **f** Time-average of the signal over the gray area in panel (**e**) vs. incident probe polarization angle $\varphi_{pr}$ (blue circles) and reference curve $\cos(2\varphi_{pr})$. The inset indicates the probe polarization rotation in the lab frame. Signals are displayed in Supplementary Note 12. The error bars are given by the standard deviation of the signal in the gray region in panel (**e**).

## Raw data

Excitation with terahertz field $+F(t)$ results in the pump-induced rotation signal $\Delta S(t,+F,\mathbf{L_0})$ (red curve in Fig. 2a), as measured for the magnetically prealigned sample with $\varphi_{pr} = \varphi_s = 0°$ (Fig. 1a). When the terahertz field is reversed, the signal $\Delta S(t,-F,\mathbf{L_0})$ (blue curve in Fig. 2a) changes sign, too, as expected for NSOTs. Therefore, we consider the signal component that is odd with respect to $F$, i.e.,

$$\Delta S(t,\mathbf{L_0}) = \frac{\Delta S(t,+F,\mathbf{L_0}) - \Delta S(t,-F,\mathbf{L_0})}{2}. \quad (1)$$

Figure 2b displays $\Delta S(t,\pm\mathbf{L_0})$ for opposite local Néel vectors. Again, the two signals are approximately reversed versions of each other, pointing to a strong contribution of the antiferromagnetic order. The signal reversal is not complete, most likely because the experimental reversal of $F$ and $\mathbf{L_0}$ has imperfections (see Supplementary Note 3). Therefore, analogous to Eq. (1), we determine signals $\Delta S(t)$, which are odd in both the pump field $F$ and the Néel vector $\mathbf{L_0}$, and focus on them in the following.

Figure 2c displays waveforms $\Delta S(t)$ from the prealigned and as-grown $Mn_2Au$ film. Remarkably, the signal from the prealigned sample is more than a factor of 5 larger and independent of the probed spot over areas much larger than the individual antiferromagnetic domains. In contrast, $\Delta S(t)$ from the as-grown sample is typically within the experimental

noise floor (Supplementary Note 4) and only exceptionally large for the example waveform depicted in Fig. 2c. For the bare $Al_2O_3$ substrate, the signal odd in $F$ is zero within the experimental accuracy. The different response of the as-grown and prealigned sample (Fig. 2c) and the very similar response of the prealigned and $Mn_2Au|Py$ test sample (Supplementary Note 5) provide strong evidence that the component $\Delta S(t)$ is related to the antiferromagnetic order of $Mn_2Au$.

Figure 2d shows that the absolute maximum of $\Delta S(t)$ grows roughly linearly with the peak amplitude $E_{pk}$ of the terahertz electric field inside the sample over the full range of $E_{pk}$. In contrast, the absolute minimum of $\Delta S(t)$ features an onset of nonlinear behavior for $E_{pk} > 30$ kV cm⁻¹ (see black-dotted lines and Supplementary Note 6). Based on measurements on the $Mn_2Au|Py$ test samples, we can exclude that the terahertz pump pulses induce magnetic switching of the antiferromagnet (Supplementary Note 5). Therefore, the signals in Fig. 2b are the linear response to the driving terahertz field ($|E| < 15$ kV cm⁻¹). We first focus on this lowest-order perturbation regime.

## Signal phenomenology

To understand the pump-probe signal $\Delta S$, we need to relate it to the perturbing terahertz pump field $F = (\mathbf{E},\mathbf{B})$ and the instantaneous magnetic order, which is quantified by $\mathbf{L}$ and $\mathbf{M}$. The symmetry properties of $Mn_2Au$ impose strict conditions on the relationship between $\Delta S$ and $F$, $\mathbf{L}$, $\mathbf{M}$, as detailed in the Methods. In particular, because the inversion

symmetry of Mn$_2$Au is broken solely by its magnetic order, pump-probe signals as those in Fig. 2c, which are linear in the driving terahertz field and odd in $\mathbf{L}_0$, are driven by the terahertz electric field $\mathbf{E}$, yet not the magnetic field $\mathbf{B}$.

Generally, the pump-probe signal depends on pump-induced changes in $\mathbf{L}$, $\mathbf{M}$ and the non-spin degrees of freedom $\mathcal{N}$. Our experimental geometry (Fig. 1a) and the point-symmetry group[31] of Mn$_2$Au imply that the pump-induced signal is up to second order in $\mathbf{L}$ and $\mathbf{M}$ given by the form

$$\Delta S \propto a \sin\left(2\varphi_{\mathrm{pr}} - 2\varphi_{\mathrm{s}}\right) \Delta\left\langle \mathbf{L}_\parallel^2 \cos\left(2\varphi_{\mathbf{L}}\right)\right\rangle \\ + b \cos\left(2\varphi_{\mathrm{pr}} - 2\varphi_{\mathrm{s}}\right) \Delta\left\langle \mathbf{L}_\parallel^2 \sin\left(2\varphi_{\mathbf{L}}\right)\right\rangle + c\mathbf{u}_z \cdot \langle\Delta\mathbf{M}\rangle + \Delta S_{\mathcal{N}}. \quad (2)$$

Here, $a$, $b$ and $c$ are sample- and setup-dependent coefficients, $\varphi_{\mathrm{pr}} - \varphi_{\mathrm{s}} = \sphericalangle(\mathbf{E}_{\mathrm{pr}}, \mathbf{u}_1)$ (Fig. 1a), and $\mathbf{L}_\parallel = \mathbf{L} - (\mathbf{u}_z \cdot \mathbf{L})\mathbf{u}_z$ is the in-plane projection of the Néel vector with azimuthal angle $\varphi_{\mathbf{L}} = \sphericalangle(\mathbf{L}_\parallel, \mathbf{u}_1)$. A preceding $\Delta$ denotes pump-induced changes, and $\langle.\rangle$ means spatial averaging over the probed Mn$_2$Au volume. The $a$ and $b$ terms of Eq. (2) are quadratic in $\mathbf{L}_\parallel$ (magnetic linear birefringence[28,33]) and monitor spin dynamics through changes in $\mathbf{L}_\parallel^2$ and $\varphi_{\mathbf{L}}$. The $c$ term is linear in $\mathbf{M}$ (magnetic circular birefringence[28]) and reports on out-of-plane variations of $\mathbf{M}$.

Note that the last term $\Delta S_{\mathcal{N}}$ of Eq. (2) is unrelated to transient changes in $\mathbf{L}$ and $\mathbf{M}$. It exclusively arises from variations of the non-spin degrees of freedom $\mathcal{N}$, such as phonons, and can be shown to exhibit the same dependence on $\varphi_{\mathbf{E}}$, $\varphi_{\mathbf{L}_0}$ and $\varphi_{\mathrm{s}}$ as the first 3 terms (see Methods). Nonmagnetic contributions of this kind are rarely considered[37]. However, Supplementary Note 7 shows that $\Delta S_{\mathcal{N}}$ makes a minor contribution to our total signal $\Delta S(t)$.

## Dominant terms

To identify the dominant terms in Eq. (2), we vary the incoming probe polarization $\varphi_{\mathrm{pr}}$ while keeping $\varphi_{\mathrm{s}} = 0°$ (Fig. 2e). We find that the waveforms $\Delta S(t, \mathbf{L}_0)$ [Eq. (1)] exhibit opposite sign for $\varphi_{\mathrm{pr}} = 0°$ and $90°$ (Fig. 2e). Their amplitude follows a $\cos(2\varphi_{\mathrm{pr}})$ dependence to very good approximation (Fig. 2f). Therefore, the signal $\Delta S(t, \mathbf{L}_0)$ predominantly derives from the $b$ term of Eq. (2). Assuming the pump-induced changes in $\mathbf{L}_\parallel^2$ and $\varphi_{\mathbf{L}}$ are small, we linearize this term. In the prealigned sample, we have $\sin(2\varphi_{\mathbf{L}_0}) = 0$ for both the 0° and 180° domains, and Eq. (2) simplifies to

$$\Delta S(t) \propto \mathbf{L}_{\parallel 0}^2 \langle\Delta\varphi_{\mathbf{L}}(t)\rangle. \quad (3)$$

This important result implies that the signal $\Delta S$ directly monitors the dynamics of the spatially averaged azimuthal rotation $\Delta\varphi_{\mathbf{L}}$ of $\mathbf{L}$, with $\Delta\varphi_{\mathbf{L}}$ being odd in $\mathbf{L}_0$ and odd in $\mathbf{E}$.

Note that a nonzero $\langle\Delta\varphi_{\mathbf{L}}(t)\rangle$ requires a non-vanishing average Néel vector $\langle\mathbf{L}_0\rangle$ in the probed volume. We can understand the occurrence of $\langle\mathbf{L}_0\rangle \neq 0$ by the magnetic prealignment procedure (see Methods and Supplementary Note 4).

## NSOT-driven terahertz magnon

As concluded above, the signal $\Delta S(t)$ cannot arise from the terahertz magnetic field. Therefore, we can discard Zeeman torque ($\propto \mathbf{B}$) and field-derivative torque[38,39] ($\propto \partial\mathbf{B}/\partial t$) as possible microscopic mechanisms. Effects of Joule heating are excluded, too, as they would scale quadratically with the pump field.

Because the Néel-vector rotation $\Delta\varphi_{\mathbf{L}}$ is linear in the terahertz electric field $\mathbf{E}$, it may arise from bulk NSOTs. To put this conjecture to the test, we consider the effect of NSOTs on the magnetic order of Mn$_2$Au. As detailed in Fig. 3a for a single $\mathbf{L}_0$ domain and a $\delta(t)$-like electric field, NSOTs induce an in-plane deflection $\Delta\varphi_{\mathbf{L}} \propto \mathbf{u}_z \cdot (\mathbf{L}_0 \times \Delta\mathbf{L})$ of the Néel vector. Because the NSOT-induced change $\Delta\mathbf{L}$ is even in $\mathbf{L}_0$ (Fig. 3a), $\Delta\varphi_{\mathbf{L}}$ is odd in $\mathbf{L}_0$, precisely as $\Delta S(t)$ [Eq. (3)]. The amplitude of $\Delta\varphi_{\mathbf{L}}$ is proportional to $\cos\sphericalangle(\mathbf{E}, \mathbf{L}_0)$. To test whether the measured

$\Delta S(t)$ follows the same dependence, we rotate the sample and, thus, the Néel vector $\mathbf{L}_0$ (Fig. 1a). We find that $\Delta S(t)$ indeed scales with $\cos\sphericalangle(\mathbf{E}, \mathbf{L}_0)$, as shown in Fig. 3b, c.

To summarize, the signal $\Delta S(t) \propto \langle\Delta\varphi_{\mathbf{L}}(t)\rangle$ is fully consistent with the terahertz-NSOTs scenario of Fig. 3a owing to its phenomenology: linear in $\mathbf{E}$, odd in $\mathbf{L}_0$ and amplitude scaling with $\cos\sphericalangle(\mathbf{E}, \mathbf{L}_0)$. The resulting predominant in-plane motion of $\mathbf{L} = \mathbf{L}_0 + \Delta\mathbf{L}$ corresponds to the in-plane magnon mode[32,40] of Mn$_2$Au. It is accompanied by an out-of-plane magnetization[40,41] $|\Delta\mathbf{M}| \propto |\partial\Delta\mathbf{L}/\partial t|$ (Fig. 3a), which is more than 2 orders of magnitude smaller than $\Delta\mathbf{L}$ and below our detection sensitivity. The remaining second magnon mode[32,40] of Mn$_2$Au, which involves an out-of-plane oscillation of $\mathbf{L}$, would be even in $\mathbf{L}_0$ and possibly be masked by contributions independent of magnetic order (Supplementary Note 8).

## Micromagnetic model

From Fig. 3a and a $\delta(t)$-like electric field, we expect a harmonic time-dependence of $\Delta\varphi_{\mathbf{L}}$ that starts sine-like. Indeed, as detailed in the Methods, the rotation $\Delta\varphi_{\mathbf{L}}$ can be described as deflection of a damped oscillator[40,42] whose potential energy $W_{\mathrm{ani}} \propto -B_{\mathrm{ani}}B_{\mathrm{ex}}\cos(4\varphi_{\mathbf{L}})$ (Fig. 4a) is determined by the in-plane anisotropy field $B_{\mathrm{ani}}$ and exchange field $B_{\mathrm{ex}}$ of the Mn$_2$Au thin film. The damping of the oscillator is proportional to $B_{\mathrm{ex}}$ and the Gilbert parameter $\alpha_{\mathrm{G}}$. The driving force scales with $\lambda_{\mathrm{NSOT}}\sigma E(t)$, where $\lambda_{\mathrm{NSOT}}$ is the NSOT coupling strength (or torkance), and $\sigma \approx 1.5$ MS m$^{-1}$ is the measured terahertz conductivity (Supplementary Note 9).

At times $t < 0$, we have equilibrium with $\varphi_{\mathbf{L}} = 0°$, 90°, 180° or 270° (Fig. 4a). Following an impulsive electric field $A\delta(t)$ with small amplitude $A$, the response $\Delta\varphi_{\mathbf{L}}$ is linear and yields a damped sinusoidal oscillation[40] $H(t) \propto \Theta(t)\mathrm{e}^{-\Gamma t}\sin(\Omega t)$ with the Heaviside step function $\Theta(t)$, frequency $\Omega/2\pi = \sqrt{\Omega_0^2 - \Gamma^2}/2\pi$, bare resonance frequency $\Omega_0/2\pi$ and damping rate $\Gamma$. For an arbitrary driving field $E(t)$, the solution of the linearized equation of motion [Eq. (17)] is given by the superposition or convolution

$$\Delta\varphi_{\mathbf{L}}(t) = (H * E)(t) = \int \mathrm{d}\tau\, E(\tau)H(t - \tau). \quad (4)$$

Figure 3d displays the measured $E(t)$ and $\Delta S(t) \propto \langle\Delta\varphi_{\mathbf{L}}(t)\rangle$ along with a fit by Eq. (4), where $\Omega_0$, $\Gamma$ and a global amplitude factor are free parameters. Equation (4) provides an excellent description for $\Omega_0/2\pi = (0.6 \pm 0.1)$ THz and $\Gamma/2\pi = (0.30 \pm 0.05)$ THz. The resulting modeled oscillatory impulse response function $H(t)$ (Fig. 3e) is strongly damped and decays within about 2 oscillation cycles.

As a cross-check, we solve Eq. (4) for $H(t)$ by numerical deconvolution without model assumptions. We find that the deconvoluted response agrees well with the fit-based result in both the time (Fig. 3e) and frequency domain (Fig. 3f). The amplitude spectrum of the transient deflection $\Delta\varphi_{\mathbf{L}}$ illustrates the broad resonance-like response given by $H$ (Fig. 3f).

Our experimentally obtained magnon frequency lies outside the accessible range of previous Brillouin- and Raman-scattering measurements[32]. A peak at 0.12 THz was ascribed to the in-plane magnon mode, but the magnetic origin of this feature is not confirmed. The bare magnon frequency $\Omega_0/2\pi$ allows us to estimate the spin-flop field by $B_{\mathrm{sf}} \sim \Omega_0/\gamma$, where $\gamma$ is the gyromagnetic ratio (see Methods). We infer $B_{\mathrm{sf}} \sim 20$ T at a temperature of 300 K, which is consistent with a previous order-of-magnitude estimate[24,43] of 30 T at 4 K. A more accurate comparison requires a detailed understanding of the magnetic reordering processes at high magnetic fields, which remain elusive. With[31] $B_{\mathrm{ex}} = 1300$ T, we extract a Gilbert-damping parameter of $\alpha_{\mathrm{G}} = \Gamma/\gamma B_{\mathrm{ex}} = 0.008$, which is consistent with theoretical predictions for metallic antiferromagnets[44,45] and recent studies[46] of optically driven spin dynamics in IrMn.

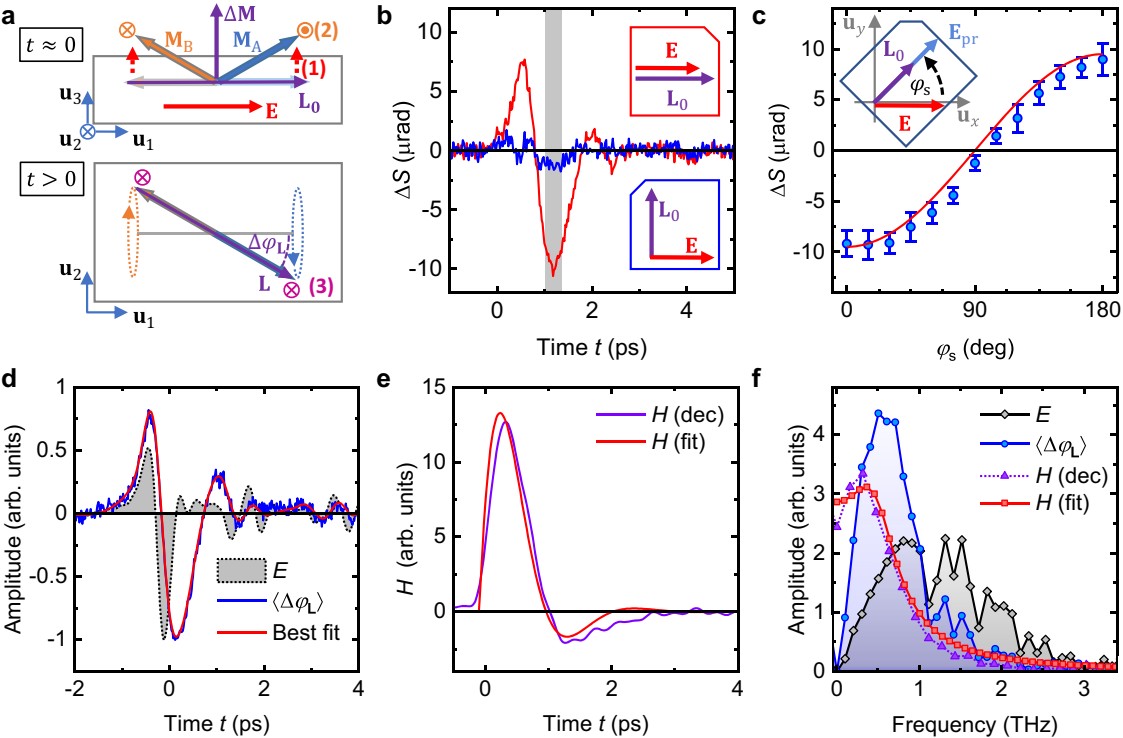

**Fig. 3 | Terahertz antiferromagnetic magnon driven by NSOTs. a** Expected dynamics driven by terahertz NSOTs, showing Mn sublattice spins A and B (orange and blue arrows). In step (1), an impulsive electric field $\mathbf{E}(t) = \mathbf{A}\delta(t)$ with strength **A** induces staggered spin-orbit fields $\propto \pm \mathbf{u}_z \times \mathbf{A}$ at A, B. Owing to $\mathbf{M}_{A0} = -\mathbf{M}_{B0} = \mathbf{L}_0/2$, the field-like torques (red-dashed lines) on A, B are equal and $\propto \mathbf{L}_0 \times (\mathbf{u}_z \times \mathbf{E})$. They cant $\mathbf{M}_A$, $\mathbf{M}_B$ and induce an out-of-plane magnetization $\Delta\mathbf{M} \propto \mathbf{L}_0 \times (\mathbf{u}_z \times \mathbf{A})$ at time $t = 0^+$. (2) Subsequently, the exchange field $B_{ex}\mathbf{L}_0/|\mathbf{L}_0|$ gradually deflects the Néel vector by $\Delta\mathbf{L} \propto (\mathbf{L}_0/|\mathbf{L}_0|) \times \Delta\mathbf{M}$ (orange symbols). (3) The anisotropy fields induce torques $\parallel \mathbf{u}_3$ (magenta symbols) and, thus, precession of $\mathbf{M}_A$ and $\mathbf{M}_B$ (dotted lines). The resulting dynamic deflection $\Delta\mathbf{L} \propto \Delta\varphi_L$ lies within the plane $\perp \mathbf{u}_3$ and fulfills $|\Delta\mathbf{L}| \gg |\Delta\mathbf{M}|$. **b** $\Delta S(t) \propto \langle\Delta\varphi_L(t)\rangle$ for $\varphi_s = 0°$ (red) and $90°$ (blue) with $E_{pk} = 15$ kV cm$^{-1}$. The probe polarization is rotated together with the sample such that $\varphi_{pr} = \varphi_s$. The insets show the relative orientation of **E**

and $\mathbf{L}_0$. **c** Time-average of the signal over the gray area in panel (**c**) vs. $\varphi_s$ (blue circles). The $\cos\varphi_s$ curve (red line) is expected from NSOTs. The inset schematically shows the simultaneous rotation of sample and probe polarization to maintain $\varphi_{pr} = \varphi_s$. The underlying signals are displayed in Supplementary Note 12. The error bars are given by the standard deviation of the signal in the gray region in panel (**b**). **d** Volume-averaged in-plane deflection angle $\langle\Delta\varphi_L(t)\rangle \propto \Delta S(t)$ of the Néel vector (blue) and the driving terahertz electric field $E(t)$ (gray-shaded area) with $E_{pk} = 15$ kV cm$^{-1}$. The red curve is the best fit of Eq. (4) to $\Delta S(t)$. **e** Time-domain impulse response $H(t)$ of $\Delta\varphi_L$ as obtained from fitting [panel (**d**), red line] or by model-free deconvolution of Eq. (4) (purple line). **f** Fourier amplitude spectra of $E(t)$ (gray diamonds) and $\langle\Delta\varphi_L(t)\rangle$ (blue circles) of panel (**d**) and of the extracted impulse response functions $H(t)$ of panel (**e**) (red squares, purple triangles).

## Nonlinear regime and torkance

Finally, we increase the peak terahertz field $E_{pk}$ inside the sample above 30 kV cm$^{-1}$ to study the nonlinear response of Mn$_2$Au indicated by Fig. 2d. Examples of signal waveforms $\Delta S(t)$, normalized to $E_{pk}$, are shown in Fig. 4b. Remarkably, as $E_{pk}$ grows (red to orange curve), the normalized signal not only decreases its peak value but also changes shape since the negative amplitude decreases relative to the positive amplitude.

Qualitatively, this transition from a linear to nonlinear response can be well understood by the anharmonic potential $W_{ani}$ of Fig. 4a and the magneto-optic detection [Eq. (2)]. First, as $W_{ani}$ grows sub-quadratically for deflection angles $\Delta\varphi_L > 15°$ (Fig. 4a), smaller restoring forces and, thus, slower dynamics than in the harmonic case result. Importantly, in the anharmonic regime, the temporal waveform is unambiguously connected to the excitation strength, as illustrated for the impulsive driving forces in Fig. 4c. Second, as the magneto-optic signal is governed by the $b$ term of Eq. (2), it exhibits a sub-linear growth at large deflection angles $\Delta\varphi_L$.

Quantitatively, we fit the measured signals (Fig. 4b) by $\Delta S(t) = C \sin[2\Delta\varphi_L(t)]$ [Eq. (16)], where $\Delta\varphi_L(t)$ is determined by numerically solving Eq. (17) of our micromagnetic model (see Methods). The only free fit parameters are the torkance $\lambda_{NSOT}$ [Eq. (17)] and the unknown prefactor $C$, which summarizes the scaling factor of our probe detection, the magneto-optic coefficient $b$ and the imbalance

between 0° and 180° domains [Eq. (16)]. We simultaneously fit all 3 curves of Fig. 4b with the same parameters and obtain good agreement of experiment and theory.

The inferred $\lambda_{NSOT} = (150 \pm 50)$ cm$^2$ A$^{-1}$ s$^{-1}$ is, to our knowledge, a first experimental determination of an NSOT torkance. It corresponds to a staggered field of $(8 \pm 3)$ mT per $10^7$ A cm$^{-2}$ driving current density, which agrees well with ab initio calculations[11] that found 2 mT per $10^7$ A cm$^{-2}$. Our procedure also allows us to calculate the signal amplitudes vs. the terahertz peak field $E_{pk}$. As seen in Fig. 2d, the onset of non-linearity is in good agreement with our experiment.

The calculated Néel-vector dynamics $\Delta\varphi_L(t)$ of a single domain are shown in Fig. 4d. We find that the rotation angle $\Delta\varphi_L$ reaches 30° at the maximum terahertz peak field of 40 kV cm$^{-1}$ inside the sample. This massive deflection is about 2 orders of magnitude larger than the magnetization deflection in ferromagnets that were induced by incident terahertz pulses comparable to ours[47,48]. It illustrates the benefits of NSOTs and exchange enhancement of the resonant Néel vector response in antiferromagnets.

## Extrapolation to switching

Our modeling enables us to extrapolate the dynamics to even higher driving fields (Fig. 4d). Remarkably, at a peak amplitude exceeding our maximum available incident field of 600 kV cm$^{-1}$ by only a factor of 2.5, the Néel vector overcomes the potential-energy maximum of the

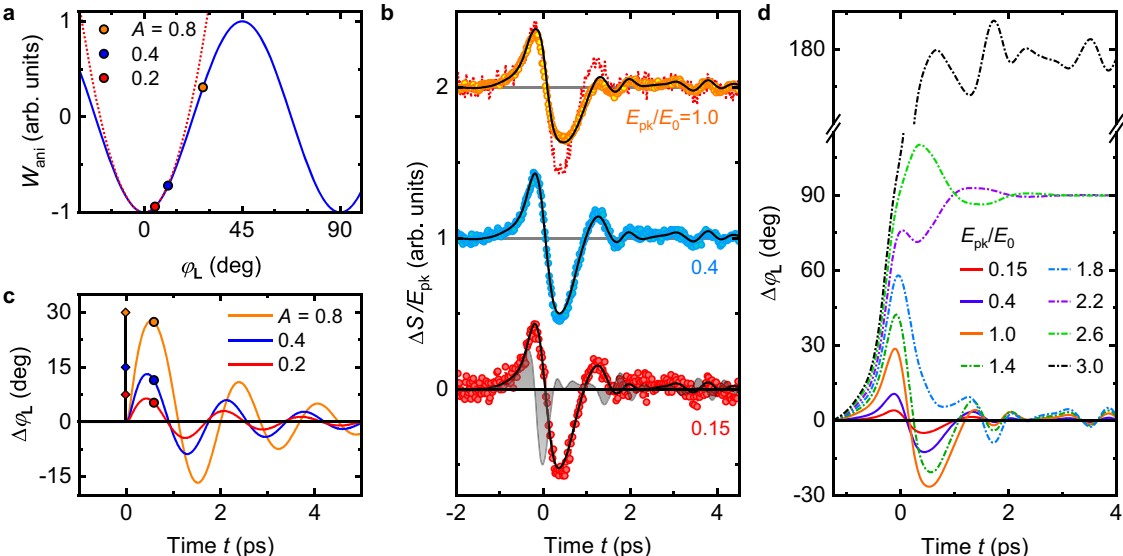

**Fig. 4 | Nonlinear Néel-vector precession and extrapolation to switching.**
**a** Magnetic anisotropy energy $W_{ani} \propto -B_{ani}B_{ex}\cos(4\varphi_L)$ vs. azimuthal rotation $\varphi_L$ of the Néel vector (blue solid line, Fig. 3a) and its harmonic approximation (red dotted line). The red, blue and orange dot indicates, respectively, the simulated position and anisotropy energy at $t = 0.6$ ps following impulsive excitation by $E(t) \propto A\delta(t)$ with relative strength $A = 0.2$, 0.4 and 0.8. **b** Normalized measured signals $\Delta S(t)/E_{pk}$ for normalized terahertz peak fields $E_{pk}/E_0$, where $E_0 = 40$ kV cm$^{-1}$ is the maximum available field inside the sample. Signals are vertically offset for clarity. The gray-shaded area shows the driving field $E(t)$ for

reference. Black lines are a joint fit using the model of Eq. (17). The red dotted line shows the signal for $E_{pk}/E_0 = 0.15$ for comparison. **c** Calculated dynamics $\Delta\varphi_L(t)$ for the 3 excitation amplitudes $A$ (diamonds) in panel (**a**). Dots correspond to $\Delta\varphi_L(t)$ at $t = 0.6$ ps [see panel (**a**)]. For illustrative purposes, the Gilbert damping parameter is chosen smaller than the experimental value. **d** Simulated dynamics of the deflection angle $\Delta\varphi_L$ for various driving fields $E_{pk}/E_0$. Solid lines correspond to the fits from panel (**b**) ($E_{pk}/E_0 \leq 1$), whereas dash-dotted lines are simulations for moderately larger fields ($E_{pk}/E_0 > 1$).

magnetic anisotropy at $\varphi_L = 45°$ (Fig. 4a) and coherently switches from $\varphi_L = 0°$ to 90°. The ultrafast switching time of only 1 ps is given by half the period $\pi/\Omega$ of the terahertz magnon. We estimate that the temperature increase due to terahertz pulse absorption is less than 5 K (Supplementary Note 10). Therefore, resonant NSOTs induced by terahertz electric pulses are an ideal driver to achieve coherent ultrafast and energy-efficient antiferromagnetic switching. When the terahertz field is increased by a factor of 3, we even obtain switching to $\varphi_L = 180°$ (Fig. 4d).

In conclusion, we show that terahertz electric fields exert field-like NSOTs in Mn$_2$Au antiferromagnetic thin films. Probing by magnetic linear birefringence reveals a strongly damped precession of the Néel vector in the sample plane at 0.6 THz. Our interpretation is consistent with regard to the symmetry and dynamics of our signals as well as their dependence on the terahertz field amplitude and magnetic-domain structure of our samples. In particular, the torkance inferred by comparison with a spin-dynamics model agrees well with ab initio predictions.

The maximum deflection of the Néel vector **L** currently amounts to as much as 30°, showing that coherent ultrafast switching of **L** by 90° without the need for heating is within reach. In addition to larger terahertz electric-field amplitudes, such experiments require a reinitialization of the magnetic state of Mn$_2$Au. The latter can potentially be achieved by an exchange-coupled ferromagnet such as Py in our test samples[29].

Our study has profound implications for future antiferromagnetic memory applications at high speeds and minimized energy consumption and might even serve as a blueprint for similar functionalities in multiferroic materials that inherently feature a linear coupling between electric and magnetic order[49]. Finally, the transient magneto-optic signals from Mn$_2$Au are significantly stronger in single-domain films (Supplementary Note 5), making them interesting candidates for spintronic detection of terahertz electromagnetic pulses with enhanced sensitivity relative to spin-orbit[50] or Zeeman torques[51].

## Methods

### Samples
The samples are epitaxial Mn$_2$Au(001) thin films (thickness of 50 nm) grown on r-cut Al$_2$O$_3$($\bar{1}\bar{1}02$) substrates (500 μm). An additional amorphous Al$_2$O$_3$ capping layer (3 nm) serves to prevent oxidation. All layers of the heterostructures are deposited by radio-frequency sputtering[52,53].

As revealed recently[15], the Mn$_2$Au films exhibit reproducible crystalline properties in the bulk. However, at the interface between Mn$_2$Au and substrate or buffer layer, a MnAu impurity phase with a thickness of roughly 10 nm is observed[15]. To keep the contribution of the interface layer to the electronic transport properties of the Mn$_2$Au thin film relatively small, we focus on samples with a ∼ 5 times larger Mn$_2$Au thickness of 50 nm. Such samples still permit a sizeable optical and terahertz transmission. Consistent with this consideration, we do not find significant NSOT-induced signals in 25 nm thick Mn$_2$Au films (Supplementary Note 11).

Following growth, one Mn$_2$Au(50 nm) sample is exposed to an intense magnetic-field pulse (peak amplitude 60 T, duration 150 ms)[24,34,35]. As a result, most domains align at 0° or 180° relative to the easy **u$_1$** axis, perpendicular to the applied field, with a ∼ 2 μm domain size as indicated in Fig. 1c.

For the control measurements (Supplementary Note 5), a stack Mo(20 nm)|Ta(13 nm)|Mn$_2$Au(50 nm)|Ni$_{80}$Fe$_{20}$(10 nm)|SiN(2 nm) is deposited on a MgO(500 μm) substrate. The Py (permalloy Ni$_{80}$Fe$_{20}$) layer is exchange-coupled to the adjacent Mn$_2$Au layer and permits control of the Mn$_2$Au antiferromagnetic order[29].

### Ultrafast setup
In our terahertz-pump magneto-optic-probe setup (Fig. 1a), the terahertz pump beam and optical probe beam are both normally incident on the sample with spot diameters of, respectively, 950 μm and 30 μm full width at half maximum (FWHM) of the intensity. The sample azimuth $\varphi_s$ is given by the angle between the **u$_1$** = [110]/$\sqrt{2}$ direction and the laboratory **u$_x$** axis.

To drive both linear and nonlinear spin dynamics, we make use of intense terahertz pump pulses (center frequency 1 THz, bandwidth $\sim$ 1 THz, field strength up to 600 kV cm$^{-1}$) generated by optical rectification[54,55] of optical femtosecond laser pulses in LiNbO$_3$ (Supplementary Note 1). The transient electric field of the terahertz pulses is characterized by electro-optic sampling[56,57] in a window of quartz (thickness of 50 μm) or ZnTe (10 μm) at the sample position analogous to the pump-probe experiment. The field strength and polarity ($\pm\mathbf{E}$) are controlled using 2 polyethylene wire-grid polarizers (Supplementary Notes 1 and 2).

The pump pulse is incident onto the metal side of the sample. Inside the Mn$_2$Au film, its electric-field amplitude amounts to about 6% of the incident amplitude (Supplementary Note 9). At 1 THz, the Al$_2$O$_3$(1$\bar{1}$02) substrate exhibits a large static birefringence with a refractive index of, respectively, 3.1 and 3.4 for the ordinary and extraordinary ray[58]. However, due to the large conductance of the metallic film, the substrate birefringence has a negligible impact on the incoupling of the terahertz electric field (Supplementary Note 9).

To monitor the pump-induced spin dynamics, we use optical probe pulses (center wavelength 800 nm, duration 20 fs, energy 0.9 nJ) from the Ti:sapphire oscillator seeding the amplified laser system[36]. They are linearly polarized with an angle $\varphi_{\mathrm{pr}} = \sphericalangle(\mathbf{E}_{\mathrm{pr}}, \mathbf{u}_x)$ relative to $\mathbf{u}_x$ (Fig. 1a) and exhibit superior power stability. Pump-induced polarization changes are detected in a balanced detection scheme, resulting in both polarization rotation and ellipticity signals $\Delta S(t)$ as a function of the pump-probe delay $t$. The birefringent Al$_2$O$_3$(1$\bar{1}$02) substrate acts as a phase retardation plate when using a probe polarization that is not along a symmetry axis of the substrate. In this case, we minimize the static birefringence contribution of the substrate by adding an identical yet 90°-rotated Al$_2$O$_3$ substrate behind it.

To identify signals related to the magnetic order of Mn$_2$Au, we measure $\Delta S(t)$ for both the local Néel vector $+\mathbf{L}_0$ and its reversed version $-\mathbf{L}_0$. While reversal of $\mathbf{L}_0$ is generally difficult, the symmetry of Mn$_2$Au permits a simple solution: Rotation of the sample by 180° about the surface normal ($\varphi_{\mathrm{s}} \rightarrow \varphi_{\mathrm{s}} + 180°$) leaves the crystal structure invariant but reverses $\mathbf{L}_0$. The sample rotation axis coincides with the optical axis of pump and probe beams within the size of the probe focus diameter and an angle of <5°. Consequently, the probed Mn$_2$Au volume remains the same.

### Probe signal

In our experiment, the linearly polarized optical probe is normally incident onto the Mn$_2$Au thin film (Fig. 1a) and induces a charge-current density $\mathbf{j}_{\mathrm{pr}}$ that is to linear order in the probe electric field $\mathbf{E}_{\mathrm{pr}}$ given by Ohm's law

$$\mathbf{j}_{\mathrm{pr}} = \underline{\sigma}\mathbf{E}_{\mathrm{pr}}. \tag{5}$$

Here, $\underline{\sigma}$ is the complex-valued second-rank conductivity tensor at the mean frequency of the probe field. The current density $\mathbf{j}_{\mathrm{pr}}$ emits a light field that is superimposed on the incident probe field and propagates to the polarimetric detection, which yields a signal

$$S \propto \left(\mathbf{E}_{\mathrm{pr0}}^* \times \mathbf{j}_{\mathrm{pr}}\right) \cdot \mathbf{u}_z. \tag{6}$$

Here, $\mathbf{u}_z$ is the sample-normal unit vector, and $\mathbf{E}_{\mathrm{pr0}}$ is the linearly polarized probe field in the absence of magnetic order and additional external fields (Fig. 1a). In this case, the sample is optically isotropic in the film plane, resulting in $\mathbf{j}_{\mathrm{pr}} \| \mathbf{E}_{\mathrm{pr0}}$ and, thus, $S = 0$. Anisotropies may lead to a nonzero signal. Note that, for the sake of simplicity, integration over all frequencies of the probe pulse and all positions of the probed volume is omitted in Eq. (6).

It is useful to consider two basis sets: the laboratory basis ($\mathbf{u}_x, \mathbf{u}_y, \mathbf{u}_z$) and the sample-fixed basis ($\mathbf{u}_1, \mathbf{u}_2, \mathbf{u}_3$), where $\mathbf{u}_1$, $\mathbf{u}_2$ and $\mathbf{u}_3$

are parallel to the crystallographic axes [110], [$\bar{1}$10] and [001] of Mn$_2$Au (Fig. 1a). In our experiment, we have $\mathbf{u}_3 = \mathbf{u}_z$, and the probe field is in the sample plane (Fig. 1a). Consequently, we can write $\mathbf{E}_{\mathrm{pr0}} = E_{\mathrm{pr1}}\mathbf{u}_1 + E_{\mathrm{pr2}}\mathbf{u}_2$ and substitute into Eq. (6). As our probe pulse is linearly polarized, we have $E_{\mathrm{pr1}}^* E_{\mathrm{pr2}} = E_{\mathrm{pr1}} E_{\mathrm{pr2}}^*$ and, thus, obtain

$$S \propto (\sigma_{22} - \sigma_{11})E_{\mathrm{pr1}}^* E_{\mathrm{pr2}} + \sigma_{\mathrm{s}}\left(|E_{\mathrm{pr1}}|^2 - |E_{\mathrm{pr2}}|^2\right) + \sigma_{\mathrm{a}}|\mathbf{E}_{\mathrm{pr0}}|^2. \tag{7}$$

Here, the conductivity tensor elements are $\sigma_{ij} = \mathbf{u}_i \cdot (\underline{\sigma}\mathbf{u}_j)$, and the symmetric and antisymmetric off-diagonal elements are, respectively, given by

$$\sigma_{\mathrm{s}} = \frac{\sigma_{21} + \sigma_{12}}{2}, \ \sigma_{\mathrm{a}} = \frac{\sigma_{21} - \sigma_{12}}{2}. \tag{8}$$

Equation (7) shows that $\sigma_{\mathrm{a}}$ induces a signal independent of the probe polarization (circular birefringence), whereas the contributions of $\sigma_{\mathrm{s}}$ and $\sigma_{22} - \sigma_{11}$ depend on the probe polarization direction (linear birefringence). Due to dissipation, the $\sigma_{ij}$ are generally complex-valued and manifest themselves in both rotation and ellipticity of polarization.

To connect $S$ to the experimentally accessible sample azimuthal angle $\varphi_{\mathrm{s}} = \sphericalangle(\mathbf{u}_1, \mathbf{u}_x)$ and probe polarization angle $\varphi_{\mathrm{pr}} = \sphericalangle(\mathbf{E}_{\mathrm{pr0}}, \mathbf{u}_x)$ (Fig. 1a), we express $E_{\mathrm{pr1}}$ and $E_{\mathrm{pr2}}$ as $|\mathbf{E}_{\mathrm{pr0}}|\cos(\varphi_{\mathrm{pr}} - \varphi_{\mathrm{s}})$ and $|\mathbf{E}_{\mathrm{pr0}}|\sin(\varphi_{\mathrm{pr}} - \varphi_{\mathrm{s}})$, respectively. As a result, Eq. (7) turns into

$$S \propto |\mathbf{E}_{\mathrm{pr0}}|^2\left[(\sigma_{22} - \sigma_{11})\sin\left(2\varphi_{\mathrm{pr}} - 2\varphi_{\mathrm{s}}\right) + \sigma_{\mathrm{s}}\cos\left(2\varphi_{\mathrm{pr}} - 2\varphi_{\mathrm{s}}\right) + \sigma_{\mathrm{a}}\right]. \tag{9}$$

The 3 terms contributing to the signal in Eq. (9) can be experimentally separated based on their different dependence on the angle $\varphi_{\mathrm{pr}} - \varphi_{\mathrm{s}} = \sphericalangle(\mathbf{E}_{\mathrm{pr0}}, \mathbf{u}_1)$ between probe field and Mn$_2$Au crystal axis $\mathbf{u}_1$, thereby yielding $\sigma_{22} - \sigma_{11}$, $\sigma_{\mathrm{s}}$ and $\sigma_{\mathrm{a}}$.

To relate $S$ to the magnetic order of the sample and the impact of the terahertz pump field ($\mathbf{E}, \mathbf{B}$), one can expand the $\sigma_{ij}$ up to linear order in the pump electromagnetic field ($\mathbf{E}, \mathbf{B}$) and up to second order in the magnetization $\mathbf{M}$ and Néel vector $\mathbf{L}$. The in-plane projection $\mathbf{L}_\| = \mathbf{L} - (\mathbf{u}_z \cdot \mathbf{L})\mathbf{u}_z$ of $\mathbf{L}$ is written as $\mathbf{L}_\| = L_\|\cos\varphi_{\mathbf{L}}\mathbf{u}_1 + L_\|\sin\varphi_{\mathbf{L}}\mathbf{u}_2$ (see Fig. 1a), and the equilibrium values of $\mathbf{M}$ and $\mathbf{L}$ are $\mathbf{M}_0 = 0$ and $\mathbf{L}_0$. The spatial symmetries of our Mn$_2$Au sample and the normally incident pump pulse greatly simplify the equations, resulting in

$$\sigma_{22} - \sigma_{11} = aL_\|^2(t)\cos\left[2\varphi_{\mathbf{L}}(t)\right] + \eta_{\mathcal{N}}L_{0\|}E(t)\sin\left(\varphi_{\mathbf{E}} - \varphi_{\mathrm{s}} + \varphi_{\mathbf{L}_0}\right),$$

$$\sigma_{\mathrm{s}} = bL_\|^2(t)\sin\left[2\varphi_{\mathbf{L}}(t)\right] + \kappa_{\mathcal{N}}L_{0\|}E(t)\cos\left(\varphi_{\mathbf{E}} - \varphi_{\mathrm{s}} + \varphi_{\mathbf{L}_0}\right),$$

$$\sigma_{\mathrm{a}} = cM_z(t) + \mu_{\mathcal{N}}L_{0\|}E(t)\cos\left(\varphi_{\mathbf{E}} - \varphi_{\mathrm{s}} - \varphi_{\mathbf{L}_0}\right). \tag{10}$$

The coefficients $a$, $b$, $c$, $\eta_{\mathcal{N}}$, $\kappa_{\mathcal{N}}$ and $\mu_{\mathcal{N}}$ are independent of $\mathbf{M}$, $\mathbf{L}$, $\mathbf{E}$ and $\mathbf{B}$. In our experiment, $\varphi_{\mathbf{L}_0}$ is an integer multiple of 90°.

The first term in each Eq. (10), which scales with $a$, $b$ or $c$, monitors true spin dynamics $\mathbf{M}(t)$ and $\mathbf{L}(t)$. In contrast, the second term in each Eq. (10), which scales with $\eta_{\mathcal{N}}$, $\kappa_{\mathcal{N}}$ or $\mu_{\mathcal{N}}$, arises from the dynamics of pump-induced changes in the non-spin degrees of freedom $\mathcal{N}$ of Mn$_2$Au. As shown in Supplementary Note 13 for each Eq. (10), the pump-induced changes in the first term have the same dependence on $\varphi_{\mathbf{E}}$, $\varphi_{\mathbf{L}_0}$ and $\varphi_{\mathrm{s}}$ as the second term. Consequently, signals due to true spin dynamics and the dynamics of non-spin degrees of freedom cannot be separated based on the variation of $\varphi_{\mathbf{E}}$, $\varphi_{\mathbf{L}_0}$ and $\varphi_{\mathrm{s}}$. Therefore, complementary arguments need to be provided, as done in Supplementary Note 7.

Note that each second term on the right-hand side of Eq. (10) is odd in both $\mathbf{E}$ and $\mathbf{L}_0$. One can generalize this property based on the fact that spatial inversion symmetry of $Mn_2Au$ is solely broken by its magnetic order (Supplementary Note 13). It follows that, in general, any signal contribution odd in $\mathbf{E}$ is simultaneously odd in $\mathbf{L}_0$ and vice versa.

We finally combine of Eqs. (9) and (10) to obtain

$$\Delta S \propto a \sin\left(2\varphi_{pr} - 2\varphi_s\right) \Delta\left[\mathbf{L}_\parallel^2 \cos\left(2\varphi_L\right)\right] \\ + b \cos\left(2\varphi_{pr} - 2\varphi_s\right) \Delta\left[\mathbf{L}_\parallel^2 \sin\left(2\varphi_L\right)\right] + c\mathbf{u}_z \cdot \Delta\mathbf{M} + \Delta S_\mathcal{N}, \quad (11)$$

where $\Delta$ indicates pump-induced changes.

### Spatial averaging of the probe

The probe signal is an integral over the probed volume. As the domains of our sample ($\sim 2\,\mu m$) are much smaller than the probe spot ($\approx 30\,\mu m$), we expect averaging over the magnetic domain configuration of the given sample. To determine the signal from a multi-domain sample, we apply spatial averaging $\langle . \rangle$ to Eq. (11) and obtain Eq. (2) of the main text. As detailed there, the observed signal symmetry allows us to write the dominant second term of Eq. (2) in its linearized form as

$$\Delta S(t) \propto b\left\langle \Delta(\mathbf{L}_{\parallel 0}^2) \sin\left(2\varphi_{L_0}\right)\right\rangle + 2b\left\langle \mathbf{L}_{\parallel 0}^2 \cos\left(2\varphi_{L_0}\right) \Delta\varphi_L\right\rangle. \quad (12)$$

In the following, we focus on the prealigned sample[24], where $\varphi_{L_0} = 0°$ and 180°. Therefore, $\sin(2\varphi_{L_0}) = 0$, $\cos(2\varphi_{L_0}) = 1$, and $\mathbf{L}_{\parallel 0}^2$ is the same for the 0° and 180° configurations, and Eq. (12) turns into Eq. (3) of the main text.

To consider the spatial averages in Eq. (2) beyond the linear limit, we note that signals $\propto \Delta(\mathbf{L}_\parallel^2)$ are negligible in the linearized case [see Eq. (12)]. We assume the same for larger signals, i.e., $\mathbf{L}_\parallel^2 \approx \mathbf{L}_{0\parallel}^2$, and obtain

$$\Delta S \propto a \sin\left(2\varphi_{pr} - 2\varphi_s\right)\left\langle \mathbf{L}_{0\parallel}^2\left[\cos\left(2\varphi_{L_0} + 2\Delta\varphi_L\right) - \cos\left(2\varphi_{L_0}\right)\right]\right\rangle \\ + b \cos\left(2\varphi_{pr} - 2\varphi_s\right)\left\langle \mathbf{L}_{0\parallel}^2 \sin\left(2\varphi_{L_0} + 2\Delta\varphi_L\right)\right\rangle + c\mathbf{u}_z \cdot \langle\Delta\mathbf{M}\rangle, \quad (13)$$

where $\langle\cdot\rangle$ denotes the average over the probed volume. As $\varphi_{L_0} = 0°$ or 180°, Eq. (13) simplifies to

$$\Delta S \propto a \sin\left(2\varphi_{pr} - 2\varphi_s\right) \mathbf{L}_{0\parallel}^2 \langle\cos(2\Delta\varphi_L) - 1\rangle \\ + b \cos\left(2\varphi_{pr} - 2\varphi_s\right) \mathbf{L}_{0\parallel}^2 \langle\sin(2\Delta\varphi_L)\rangle + c\mathbf{u}_z \cdot \langle\Delta\mathbf{M}\rangle, \quad (14)$$

where the average $\langle . \rangle$ runs over just $\varphi_{L_0} = 0°$ and 180°. For small deflections $\Delta\varphi_L$, Eq. (14) can be expanded up to quadratic order as

$$\Delta S \propto 2a \sin\left(2\varphi_{pr} - 2\varphi_s\right) \mathbf{L}_{0\parallel}^2 \langle\Delta\varphi_L^2\rangle \\ + 2b \cos\left(2\varphi_{pr} - 2\varphi_s\right) \mathbf{L}_{0\parallel}^2 \langle\Delta\varphi_L\rangle + c\mathbf{u}_z \cdot \langle\Delta\mathbf{M}\rangle. \quad (15)$$

The deflection angle $\Delta\varphi_L$ and out-of-plane magnetization $\Delta M_z = \mathbf{u}_z \cdot \Delta\mathbf{M}$ are odd in both $\mathbf{E}$ and $\mathbf{L}_0$, since $\Delta S$ is required to be odd in $\mathbf{E}$ and $\mathbf{L}_0$ by definition. It follows that $\Delta\varphi_L^{0°} = -\Delta\varphi_L^{180°}$ and $\Delta M_z^{0°} = -\Delta M_z^{180°}$, where the superscripts indicate domains with $\varphi_{L_0} = 0°$ and 180°, respectively. Assuming the $\mathbf{L}_0$-dependence of $\Delta\varphi_L^{0°}$ and $\Delta M_z^{0°}$ is linear, the spatial average of both $\Delta\varphi_L$ and $\Delta M_z$ scales with $\langle\mathbf{L}_0\rangle$ in the prealigned sample. Therefore, observation of a signal from the $\langle\Delta\varphi_L\rangle$ term of Eq. (15) requires $\langle\mathbf{L}_0\rangle \neq 0$. This contribution has a smaller amplitude than in the case of a single domain, but unaltered dynamics.

The preceding argumentation also holds for the last 2 terms in Eq. (14). More explicitly, for $\varphi_{pr} - \varphi_s = 0°$ or 180°, Eq. (14) turns into

$$\Delta S \propto \left(f^{0°} - f^{180°}\right)\left[b\mathbf{L}_{0\parallel}^2 \sin\left(2\Delta\varphi_L^{0°}\right) + c\mathbf{u}_z \cdot \Delta\mathbf{M}^{0°}\right], \quad (16)$$

where $f^{180°}$ and $f^{180°}$ denotes the volume fraction of domains with $\varphi_{L_0} = 0°$ and 180°, respectively, with $f^{0°} + f^{180°} = 1$. Importantly, Eq. (16) includes nonlinear responses odd in both $\mathbf{E}$ and $\mathbf{L}_0$.

The origin of $f_{0°} \neq f_{180°}$ and, thus, $\langle\mathbf{L}_0\rangle \neq 0$ lies in the domain reorientation by the strong magnetic-field pulse, as discussed in Supplementary Note 4.

### Micromagnetic model details

Two uniform antiferromagnetic magnon modes are predicted to exist in $Mn_2Au$. They exhibit predominantly in-plane and out-of-plane oscillation character of the Néel vector, respectively[40], as detailed in Supplementary Note 8.

The dynamics of the in-plane mode is captured by the uniform azimuthal rotation $\varphi_L(t)$ (Fig. 3a). The equation of motion for this mode in a given domain driven by NSOTs is[40,42]

$$\ddot{\varphi}_L + 2\gamma\alpha_G B_{ex}\dot{\varphi}_L + \left(\Omega_0^2/4\right)\sin(4\varphi_L) \\ = -\gamma B_{ex}\lambda_{NSOT}\sigma\left[(\mathbf{E}\cdot\mathbf{u}_1)\cos\varphi_L + (\mathbf{E}\cdot\mathbf{u}_2)\sin\varphi_L\right]. \quad (17)$$

Here, $\gamma$ is the gyromagnetic ratio, $B_{ex} \approx 1300\,T$ is the inter-sublattice exchange field[31], $\alpha_G$ is the Gilbert damping parameter, $\Omega_0^2 = 4\gamma^2 B_{ex}B_{ani}$, where $B_{ani}$ is the in-plane anisotropy field in the definition of Ref. 40, $\lambda_{NSOT}$ is the NSOT coupling strength (or torkance), and $\sigma \approx 1.5\,MS\,m^{-1}$ is the approximate measured conductivity at 0–3.5 THz. The terahertz electric field $\mathbf{E}(t)$ inside the thin film amounts to about 6% of the incident field (Supplementary Note 9).

Equation (17) has interesting properties. First, it is strongly non-linear with respect to the Néel-vector deflection $\Delta\varphi_L(t)$ (third term on the left-hand side) and, thus, the driving electric field $\mathbf{E}$. Second, Eq. (17) implies that the in-plane antiferromagnetic-mode frequency is fully determined by the in-plane anisotropy and exchange fields. The spin-flop field is connected to these quantities by[40] $B_{sf} = 2\sqrt{B_{ani}B_{ex}}$ and, thus, to the bare magnon frequency $\Omega_0/2\pi$ through $\Omega_0 \sim \gamma B_{sf}$. This relationship allows us to directly relate the measured $\Omega_0$ to results from complementary experiments[24,35].

Third, when going from one domain with equilibrium orientation $\varphi_{L_0}$ to the oppositely oriented domain with angle $\varphi_{L_0} + 180°$, the right-hand side of Eq. (17) changes sign. Therefore, the direction of rotation of $\mathbf{L}$ also changes sign, from, e.g., clockwise to counter-clockwise (Fig. 1c).

## Data availability

The data that support the plots in the main text of this paper are openly available on Zenodo under https://doi.org/10.5281/zenodo.8255849 (Ref. 59). The data that support other findings of this study are available from the corresponding author on request.

## Code availability

Custom computer codes or algorithms used to generate results that are reported in the paper and central to its main claims are available from the corresponding authors on request.

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

## Acknowledgements

We acknowledge funding by the German Research Foundation through the collaborative research centers SFB TRR 227 "Ultrafast spin dynamics" (project ID 328545488, projects A05 and B02) [Y.B., A.L.C., O.Gue., T.S.S., T.K.], SFB TRR 173 "Spin+X" (project ID 268565370, projects A01, A05, A11, B02 and B12) [S.Y.B., S.R., Y.L., O.Go., M.K., M.J.] and priority program SPP2314 INTEREST (project ITISA) [Y.B., A.L.C., O.Gue., T.S.S., T.K.], the European Union through the projects ERC H2020 CoG TERAMAG/grant no. 681917 [Y.B., A.L.C., O.Gue., T.S.S., T.K.] and ERC SyG 3D MAGiC/grant no. 856538 [S.Y.B., S.R., Y.L., M.K., M.J.]. We acknowledge KAUST (Grant No. OSR-2019-CRG8-4048) [S.Y.B., S.R., Y.L., M.K., M.J.] and support by the European Commission under FET-Open Grant Agreements No. 863155 (s-Nebula) [Y.B., A.L.C., O.Gue., T.S.S., T.K., S.Y.B., S.R., Y.L., M.K., M.J.] and No. 766566 (ASPIN) [S.Y.B., S.R., Y.L., O.Go., M.K., M.J.] and No. 101070287 (SWAN-on-chip) [M.K., M.J., S.R.].

## Author contributions

T.K., Y.B. and A.L.C. conceived the experiment. Y.B., A.L.C. and O. Gueckstock performed the experiments. Y.B., A.L.C. and O. Gueckstock analyzed the data. S.Y.B. and Y.L. prepared samples, and S.Y.B., S.R., Y.L. and M.J. characterized them. Y.S. performed the spin-flop prealignment. O. Gomonay provided the basis of the theoretical model. Y.B., A.L.C. and T.K. wrote the original draft. T.S.S., M.W., O. Gomonay, M.K., S.R. and M.J. reviewed and edited the paper, which was subsequently read and commented on by all authors.

## Funding

## Competing interests

The authors declare no competing interests.
