## [Peer Review File · Nature Communications]

Reviewers' Comments:

Reviewer #1:

Remarks to the Author:

The manuscript by Y. Behovits et al. reports on the experimental study of spin dynamics driven by picosecond THz pulse in an antiferromagnet Mn₂Au. The authors aim at exploring so-called Neel spin-orbit torques (NSOT) for excitation of coherent dynamics of spins in this antiferromagnet. The authors demonstrate linear- and nonlinear regimes of spin dynamics excited by electric field of the THz pulse and ascribe them to action of NSOT.

The subject of the manuscript is timely, as it deals with the problem of finding and exploring novel approaches to excite and potentially switch antiferromagnetic vector by electric stimuli. The manuscript contains extensive experimental results supported by detailed symmetry analysis of the suggested effect. In general, the presented results and conclusions are convincing. One could expect that the findings reported in the manuscript would be of interest to rather broad community of researchers working in spintronics and ultrafast magnetism. Thus, I can recommend the manuscript for publication.

I have, nevertheless, several comments regarding the manuscript.

1. Since the central frequency of the observed spin dynamics lies well within the spectrum of the THz pulse, why this mode is not seen in the transmission spectrum presented in the Supp. Material? I believe the comment on this should be provided, as the correct identification of the nature of the observed dynamics is essential.
2. It is not very clear from the manuscript, if the NSOT torque λ_{NSOT} was the same in simulations performed for the linear and nonlinear case. Does this value depend on E_{pk} ? In my opinion only in this case the torque can be referred to as nonlinear, as it is done in the manuscript title. To my understanding, it is the magnetization response, which is nonlinear, while the torque itself remains linear.
3. In the main text, it would be useful to provide a brief explanation why the excitation of magnetic dynamics by the magnetic field of the THz pulse can be excluded.
4. Line 264-265. The phrase is somewhat confusing: "Remarkably, as E_{pk} grows (red to orange curve), the normalized signal not only decreases its peak value but also becomes more symmetric". Probably, the authors could be more specific what is meant by "more symmetric".
5. Line 154. The reference to a wrong figure (Fig 2 instead of Fig. 1) is given.

Reviewer #2:

Remarks to the Author:

Antiferromagnetic-based spintronics devices, immune to external magnetic perturbation, have immense applications for high-speed electronics. Additionally, they can have large applications for ultrafast coherent switching of magnetic order with minimum heat dissipation. Here the authors have monitored the spin response with a femtosecond magneto-optics probe by applying phase-locked single-cycle THz pulses to an intrinsically asymmetric antiferromagnetic (Mn₂Au) thin film. The main outcomes of this manuscript are
Above a certain incident, THz electric field pronounced nonlinear dynamics with massive Vector deflection was observed.

The experimental results were backed by a micromagnetic model indicating that achieving fully coherent Néel Vector switching by 90° within 1 ps will be possible in the future.

The investigation, results, and explanation of the manuscript is concise. In my opinion, the manuscript warrants publication, but I have some comments that I want the authors to address. There are certain areas that the authors need to clarify further, and I have a few words concerning

the paper, as explained below:

Why did the authors use 50 nm of Mn₂Au for investigation? How did they optimize to reach the mentioned thickness for investigation? Can the author comment on the thickness-dependent signal response study?

In Fig 2, signal strength seems to change if we flip the pump side and L0. Can the author comment on why the slight difference in signal is happening? Especially with Fig 2(b), there should not be any change in signal strength when we rotate the sample by 180°.

The authors have used R-cut sapphire as a substrate for the measurement which is known to have anisotropic behavior in the THz region. Is the anisotropy of the substrate taken care of when reporting the results?

In Fig 2c, it is evident that $\Delta S(t)$ for the as-grown sample is very low and within the noise floor, so how did the author confirm that the signal is odd with respect to L0 and not even? What does the author mean by the statement " $\Delta S(t)$ from the as-grown sample is typically within the experimental noise floor (see Fig. S5) and only exceptionally large for the example waveform depicted in Fig. 2c" on page 7? Even in Fig. S5, the signal is almost 0 for all the probed spots; then how does the author confirm the signal is odd?

How, from the bare substrate, is the signal odd? Can the authors explain it? In my understanding, the bare sapphire should not show any changes if we rotate it by 180°. And that is evident in Fig 2C, where the $\Delta S(t)$ is 0.

Apart from these, I also have some general comments below.

Can the authors provide some evidence of the growth quality of the sample through characterization? As they mentioned Mn₂Au is an exciting material with intrinsic broken inversion symmetry, and the epitaxial growth of the sample is essential to claim anything, hence it is necessary to check and verify the growth quality of such samples.

Reviewer #3:

Remarks to the Author:

The work of Behovits and co-workers presents an experimental study of terahertz frequency pumping on the antiferromagnet Mn₂Au. There is a significant body of work here with careful checking of many of the experimental details as well as some theoretical modelling. The main outcome of the article is the demonstration of ultrafast dynamics in Mn₂Au, a material which has been extensively studied in recent years, as the authors detail. The previous studies on controlling the magnetic order have not been performed at this ultrafast timescale, therefore I agree with the authors that this is an important result for the field of ultrafast magnetism and could result in further experiments and theoretical works in other antiferromagnets. The manuscript is well organized and there is a lot of additional (and beneficial to the reader) information in the supplementary information and methods sections. It is unfortunate that the authors could not achieve experimental switching, but I think that the demonstration of the ultrafast response is sufficiently interesting to warrant publication in Nature Communications.

There are, however, a few details that I thought required clarification.

- The initial alignment of the domains in the sample by an intense magnetic field pulse which is discussed in the main article and the methods. However, the dynamics are performed in a pump-probe setup, using a stroboscopic technique. Usually, the initial state is reset as the experiment is performed in a constant magnetic field. How do the authors ensure that the same initial state is achieved?

- In relation to the previous point, if the initial state cannot be reset then how could one determine whether switching has occurred? If, for example, the size of the electric field generated could be

increased to the values predicted to result in switching (as shown in figure 4d by numerical modelling) then some domains would rotate at a certain delay and not be back switched for the next delay. The resulting stroboscopic values of Delta S might then be discontinuous when creating the time trace.

- In the paragraph starting on line 272, is the scaling factor due to the difference between what the numerical model produces, something like the angle M makes as a function of time and Delta S?

Response letter for manuscript entitled
**Nonlinear Néel spin-orbit torques in
antiferromagnetic Mn₂Au***

*Title will be modified (see C1.2)

Response to the Reviewers

We would like to thank all Reviewers for their interest in our work, the time they spent on our manuscript and their comments. We implemented all their comments, resulting in a, we think, significantly improved manuscript that warrants publication in line with the referee evaluations.

Our response and manuscript changes according to the Reviewers' suggestions are detailed further below. In the new manuscript, **significant text changes are highlighted yellow**, often followed by a code such as "**[C2.3]**", meaning that the preceding text addresses comment#3 of Reviewer#2.

Reviewer #1 (Remarks to the Author):

The manuscript by Y. Behovits et al. reports on the experimental study of spin dynamics driven by picosecond THz pulse in an antiferromagnet Mn₂Au. The authors aim at exploring so-called Neel spin-orbit torques (NSOT) for excitation of coherent dynamics of spins in this antiferromagnet. The authors demonstrate linear- and nonlinear regimes of spin dynamics excited by electric field of the THz pulse and ascribe them to action of NSOT.

The subject of the manuscript is timely, as it deals with the problem of finding and exploring novel approaches to excite and potentially switch antiferromagnetic vector by electric stimuli. The manuscript contains extensive experimental results supported by detailed symmetry analysis of the suggested effect. In general, the presented results and conclusions are convincing. One could expect that the findings reported in the manuscript would be of interest to rather broad community of researchers working in spintronics and ultrafast magnetism. Thus, I can recommend the manuscript for publication.

Response: We thank the reviewer for their detailed comments and appreciation of our work. In the following, we address all comments point-by-point.

I have, nevertheless, several comments regarding the manuscript.

C1.1: 1. Since the central frequency of the observed spin dynamics lies well within the spectrum of the THz pulse, why this mode is not seen on the transmission spectrum presented in the Supp. Material? I believe the comment on this should be provided, as the correct identification of the nature of the observed dynamics is essential.

Response: We fully agree the magnon should be visible in the THz transmission spectrum because the THz electric field transfers power to the magnon mode, which is finally dissipated by other degrees of freedom, e.g., phonons. By energy conservation, this process should leave a signature in the transmitted/reflected THz wave.

Indeed, in insulating antiferromagnets, THz time-domain spectroscopy (THz-TDS) has been used extensively to identify magnons. Examples include comparatively thick (~1mm) crystals of TmFeO₃ [1], CoF₂ [2] and MnO [3].

In contrast, we are not aware of studies where THz-TDS was used to identify magnons in metallic antiferromagnets by NSOTs. For our Mn₂Au sample, we would expect that the in-plane magnon would lead to a THz-TDS signal with the symmetry of anisotropic magnetoresistance, i.e., quadratic in the Neel vector. In ferromagnets, such signals are typically rather small and require fast modulation of the magnetic order [4].

A conservative estimate based on the inverse-NSOTs mechanism of ref. [5] leads to an amplitude modulation of the order of 1% between the configurations $\mathbf{E} \parallel \mathbf{L}_0$ and $\mathbf{E} \perp \mathbf{L}_0$. The challenge of this experiment consists of the low mode frequency (<1 THz), its large damping (spectrally broad compared to the magnons in [1-3]) and the strong attenuation of the THz field in the >50 nm thick metal stack.

In the THz transmission measurements shown in Fig. S6, we could not resolve the effect. However, we plan to study the Py-capped samples in the future, where in-situ control of the magnetic order by an external magnetic field is possible, which should ease detection of the magnon mode in THz transmission experiments.

Action: We accordingly added a line to Supplementary Note 8 (line 800), stating that the mode should be in principle observable, but is not found in our experiment.

C1.2: 2. It is not very clear from the manuscript, if the NSOT torque λ_{NSOT} was the same in simulations performed for the linear and nonlinear case. Does this value depend on E_{pk} ? In my opinion only in this case the torque can be referred to as nonlinear, as it is done in the manuscript title. To my understanding, it is the magnetization response, which is nonlinear, while the torque itself remains linear.

Response: We thank the reviewer for raising this important point. The torque is indeed the same. In principle, the only “truly” nonlinear torques that emerge are those from the anisotropy at large angles. Therefore, we fully agree that our initial title is short but may be misleading.

Action: We accordingly changed the title from

“Nonlinear terahertz Néel spin-orbit torques in antiferromagnetic Mn₂Au” to

“Terahertz Néel spin-orbit torques drive nonlinear magnon dynamics in antiferromagnetic Mn₂Au”

We also added a note to the section “Methods/Micromagnetic model details”, clarifying that the torque is unchanged and the non-linearity stems from the deflection \mathbf{L} alone (line 474).

C1.3: 3. In the main text, it would be useful to provide a brief explanation why the excitation of magnetic dynamics by the magnetic field of the THz pulse can be excluded.

Response: We thank the referee for raising this important point. Our previous argumentation on the role of the THz magnetic field \mathbf{B} (previous line 187) was obviously too terse, and we now expand accordingly.

The spatial inversion symmetry of Mn₂Au is broken only due to its magnetic order. It follows that changes in \mathbf{L} and \mathbf{M} that are (i) linear in the driving THz field and (ii) odd in the static \mathbf{L}_0 must be driven by \mathbf{E} , not \mathbf{B} . In contrast, if the observed signal was linear in \mathbf{B} , it would have to be even in \mathbf{L}_0 and, thus, not change under reversal of \mathbf{L}_0 .

We also note that interface torques can be excluded due to identical dynamics in samples with different interfaces (Supplementary Note 5).

Action: We have expanded the explanation around line 204.

C1.4: 4. Line 264-265. The phase is somewhat confusing: “Remarkably, as E_{pk} grows (red to orange curve), the normalized signal not only decreases its peak value but also becomes more symmetric”. Probably, the authors could be more specific what is meant by “more symmetric”.

Response: We thank the reviewer for pointing this out. We meant to state that the signal changes its shape since the negative amplitude decreases relative to the positive amplitude.

Action: We updated the text accordingly and added “...and changes shape since the negative amplitude decreases relative to the positive amplitude” in line 284.

C1.5: 5. Line 154. The reference to a wrong figure (Fig 2 instead of Fig. 1) is given.

Response: Fig. 2a is actually the correct reference, but we did not sufficiently clarify that we mean the waveform $E(t)$.

Action: We changed line 164 to “[...] component $\mathbf{E} = E\mathbf{u}_x$, with its waveform $E(t)$ shown in Fig. 2a”

Reviewer #2 (Remarks to the Author):

Antiferromagnetic-based spintronics devices, immune to external magnetic perturbation, have immense applications for high-speed electronics. Additionally, they can have large applications for ultrafast coherent switching of magnetic order with minimum heat dissipation. Here the authors have monitored the spin response with a femtosecond magneto-optics probe by applying phase-locked single-cycle THz pulses to an intrinsically asymmetric antiferromagnetic (Mn₂Au) thin film. The main outcomes of this manuscript are

Above a certain incident, THz electric field pronounced nonlinear dynamics with massive Vector deflection was observed.

The experimental results were backed by a micromagnetic model indicating that achieving fully coherent Néel Vector switching by 90° within 1 ps will be possible in the future.

The investigation, results, and explanation of the manuscript is concise. In my opinion, the manuscript warrants publication, but I have some comments that I want the authors to address. There are certain areas that the authors need to clarify further, and I have a few words concerning the paper, as explained below:

Response: We thank the reviewer for their detailed comments and appreciation of our work. In the following, we address all comments point-by-point.

C2.1: Why did the authors use 50 nm of Mn₂Au for investigation? How did they optimize to reach the mentioned thickness for investigation? Can the author comment on the thickness-dependent signal response study?

Response: A thickness of 50 nm provides a good compromise with regard to (i) structural quality (please see the detailed discussion in C2.6) and (ii) optical properties (i.e., sufficient coupling of THz electric field into the Mn₂Au film and sizeable transmittance of the magneto-optic probe-beam).

We also studied films with a thickness of 25 nm (see Fig. C2.1 below). We find that both as-grown and field-exposed (“prealigned”) sample show similar signals, containing both a step-like and an oscillatory response. In one of the experimental configurations (for terahertz field $+F$). As we discuss in response C2.2 and the newly added Supplementary Note 3, this step originates from the Kerr effect (transient birefringence quadratic in the driving field, e.g., in the substrate) and an imperfect reversal of the terahertz field (i.e., “ $+F$ ” and “ $-F$ ” are not exactly anti-parallel). The resulting signal effect scales with F^2 and, thus, $1/d^2$, where d is the metal film thickness (see Supplementary Note 9).

Thus, the Kerr effect is much more pronounced in thinner films, which makes it challenging to study spin dynamics there. If the signal is still present in the 25 nm sample on top of the strong Kerr response, it must be about an order of magnitude smaller than in the 50 nm samples. This notion is consistent with a larger relative role of an interfacial MnAu impurity phase (see Methods and ref. [6]), which is not expected to feature the NSOTs present in epitaxial Mn₂Au.

Action: We accordingly added a discussion to the “Methods/Samples” section (line ~340) and included Fig. C2.1 as a Fig S11 in the Supplementary Information (line ~838) along with the explanation above.

Fig. C2.1: Signals for as-grown and prealigned 25 nm films a) Signals of the as-grown film. Blue and orange lines are the signals for configurations $\pm F$, respectively. b) Signal of the prealigned film. Blue

(red) dotted lines are raw data for configuration $\pm F$, whereas blue (red) solid lines are smoothed. The lower signal-to-noise ratio compared to a) stems from a 10 times shorter averaging time in these measurements.

C2.2: In Fig 2, signal strength seems to change if we flip the pump side and L_0 . Can the author comment on why the slight difference in signal is happening? Especially with Fig 2(b), there should not be any change in signal strength when we rotate the sample by 180° .

Response: The reviewer points out the shortcomings of our separation technique, which is based on flipping the polarity of the THz pump pulse (shorthand $+F \rightarrow -F$) and rotating the sample azimuthally by 180° (shorthand $+L_0 \rightarrow -L_0$). This procedure is sensitive to a number of degrees of freedom in alignment.

For example, the electric fields generated by the rotating polarizers are not perfectly antiparallel, but contain an additional component, due to the transmission through non-ideal polarizers. This can lead to a signal contribution $\propto E_x E_y$ (e.g., electro-optical Kerr effect in the substrate) with different amplitude for the two field directions.

As shown in the attached Fig. C2.2a (showing the raw data for manuscript Fig. 2), when changing from $-F, +L_0$ to $-F, -L_0$, the waveforms are almost perfectly reversed. However, going from $+F, +L_0$ to $+F, -L_0$, there is a significant change of the waveform in both shape and amplitude. Thus, for $+F$, we observe a signal even in L_0 (i.e. sample rotation). We attribute this to the electro-optical Kerr effect being mainly present in configuration $+F$, but not in $-F$.

We explain it such that, the electric field present for $+F$ is approximated by $E\mathbf{u}_x + \epsilon\mathbf{u}_y$, whereas for $-F$, it is simply $-E\mathbf{u}_x$. This leads to a Kerr signal $\propto E\epsilon$, which is apparently odd in F , yet even in L_0 , since it does not depend on L_0 at all.

Fig C2.2b shows all signal combinations. Besides the dominant F^1, L_0^1 -component, there are two minor contributions even in L_0 . The latter are thus effectively removed from the relevant signal by the double-modulation of both F and L_0 .

We realized that the field reversal measured by electro-optical sampling shown in Fig. S2 cannot be called “nearly fully reversing”. This is only true for the field component parallel to the sensitive axis of the electro-optical crystal, not for the one perpendicular to it (which is not measured there). We clarified this in the Supplementary Note 2.

Action: We added text on this issue around lines 171 and 180. Further, we clarified Supplementary Note 2 by stating that the field reversal is only verified along one polarization axis (line 553, 571). We also added the data for all combinations $\pm F$ and $\pm L_0$ in Supplementary Note 3 (line 577).

Fig. C2.2: a) Raw signals of the 4 experimental geometries. The signals for $+F, -L_0$ and $-F, +L_0$ have been multiplied by -1 for better comparison. b) Signal contributions odd and even in F and L_0 .

C2.3: The authors have used R-cut sapphire as a substrate for the measurement which is known to have anisotropic behavior in the THz region. Is the anisotropy of the substrate taken care of when reporting the results?

Response: The reviewer makes an important point since the anisotropy of the R-cut sapphire substrate may affect both the THz pump and the optical probe pulse. To mitigate this effect, pump and probe traverse the sample in the sequence metal and subsequently sapphire (Al₂O₃).

With regard to the terahertz pump pulse, the refractive index of Al₂O₃ is $n_o = 3.1$ for the ordinary ray and $n_e = 3.4$ for the extraordinary ray [7] at a frequency of 1 THz. Even though this birefringence is considerable, it is still negligible because the THz-field incoupling is dominated by the conductance of the metal film. This effect is readily seen in Eqs. (18) and (20) (Supplementary Material) where the conductance term in the denominator is $Z_0 G \approx 30$ is $\gg n_o, n_e$. The relative difference of the ordinary and extraordinary THz electric field inside the metal film is, thus, on the order of 10^{-2} .

With regard to the femtosecond probe pulse, the birefringence is smaller ($(n_e - n_o)/n_o < 1\%$) than at 1 THz (10%). However, the probe pulse will acquire additional ellipticity if its polarization is not along a high symmetry axis of the substrate as in Fig. 2e,f. To compensate for this effect, we use another, 90° rotated substrate when measuring the probe polarization dependence in Fig 2e,f. The sample is, therefore, metal|sapphire1|sapphire2, where sapphire2 is rotated by 90° relative to sapphire1.

Action: We added clarifications to the “Methods (line 366) and Supp. Note 9 (line 783 and 799)

C2.4: In Fig 2c, it is evident that $\Delta S(t)$ for the as-grown sample is very low and within the noise floor, so how did the author confirm that the signal is odd with respect to L0 and not even? What does the author mean by the statement “ $\Delta S(t)$ from the as-grown sample is typically within the experimental noise floor (see Fig. S5) and only exceptionally large for the example waveform depicted in Fig. 2c” on page 7? Even in Fig. S5, the signal is almost 0 for all the probed spots; then how does the author confirm the signal is odd?

Response: We thank the reviewer for pointing out our slightly ambiguous description in the caption.

For the Mn2Au samples, we simply take the difference signal for the sample at 0° and 180° azimuth. According to the symmetry properties of Mn2Au, this rotation implies that the local \mathbf{L}_0 turns into $-\mathbf{L}_0$. If any mean $\langle \mathbf{L}_0 \rangle$ is present, it rotates in the same manner. If $\langle \mathbf{L}_0 \rangle = 0$, taking this difference one obtains zero signal.

In contrast to the prealigned sample, the difference signal from as-grown sample does not show a sizeable contribution, indicating that $\langle \mathbf{L}_0 \rangle$ is much smaller than in the prealigned sample (possibly zero). Please note that Fig.2c shows the maximum signal we found in the as-grown sample (also see Fig.S5).

Action: We accordingly changed the label in Fig.2, added text (line ~55) and also clarified this in the description of the experiment (line ~171).

C2.5: How, from the bare substrate, is the signal odd? Can the authors explain it? In my understanding, the bare sapphire should not show any changes if we rotate it by 180°. And that is evident in Fig 2C, where the $\Delta S(t)$ is 0.

Response: The reviewer is completely right. The substrate signal is indeed shown for just for one orientation. Since there is no discernible signal, i.e. $\Delta S(t, F^1) = 0$, we do not compare the substrate at angle 0° and at 180°.

Action: We clarified this in the caption (line ~55) and figure axis label.

Apart from these, I also have some general comments below.

C2.6: Can the authors provide some evidence of the growth quality of the sample through characterization? As they mentioned Mn2Au is an exciting material with intrinsic broken inversion symmetry, and the epitaxial growth of the sample is essential to claim anything, hence it is necessary to check and verify the growth quality of such samples.

Response: We fully agree that more information should be added.

The sample growth by sputtering results in highly reproducible crystallographic properties. More precisely, the samples used for this work were grown with the same parameters as those from Ref. [8], in which epitaxial growth was confirmed by X-ray diffraction (XRD).

In a more recent work [6] (in particular Fig. S3), we identified a MnAu impurity phase that is present in a layer with a thickness of roughly 10 nm at the interface of the Mn₂Au thin film and the Ta buffer layer. This phase corresponds to the XRD peak labeled (101) in Fig. 1 of Ref. 50 [8].

Note that the MnAu-phase layer appears in XRD exactly in the same way for all Mn₂Au(001) thin films, independent of the specific substrate and buffer layer used. In Fig. C2.6, the XRD data for two films grown with and without Ta buffer layer is shown. The film without Ta is grown with the same parameters as the samples that are studied in the pump-probe experiments. Furthermore, for Mn₂Au film thicknesses smaller than 20 nm, a broadening of the XRD rocking curves is observed. We conclude that the impurity phase is always forming at the interface with the substrate or buffer layer.

To keep the relative contribution of this impurity phase to the electronic transport properties of the Mn₂Au thin film negligible, we chose a layer thickness which is ~5 times larger than the MnAu layer.

Fig. C2.6: XRD $\theta/2\theta$ -scans of an Al₂O₃(1102)/Ta(100),13 nm/Mn₂Au(001),40 nm(black curve) and Al₂O₃(1102)/Mn₂Au(001),50 nm(red curve) thin films grown on Al₂O₃(1102) substrates. The sample without the Ta buffer layer was grown at T=720 °C and post-annealed at T= 700 °C for 1h 15 mins, and the sample with the Ta buffer layer was grown at T=500 °C and post-annealed at T=700 °C for 1h 15 mins. Both thin films show very similar intensity of the XRD peaks which reflects their similar crystal quality.

Action: We accordingly added text to the sections “Experiment” (line ~152) and “Methods/Samples” (line ~340).

Reviewer #3 (Remarks to the Author):

The work of Behovits and co-workers presents an experimental study of terahertz frequency pumping on the antiferromagnet Mn₂Au. There is a significant body of work here with careful checking of many of the experimental details as well as some theoretical modelling. The main outcome of the article is the demonstration of ultrafast dynamics in Mn₂Au, a material which has been extensively studied in recent years, as the authors detail. The previous studies on controlling the magnetic order have not been performed at this ultrafast timescale, therefore I agree with the authors that this is an important result for the field of ultrafast magnetism and could result in further experiments and theoretical works in other antiferromagnets. The manuscript is well organized and there is a lot of additional (and beneficial to the reader) information in the supplementary information and methods sections.

It is unfortunate that the authors could not achieve experimental switching, but I think that the demonstration of the ultrafast response is sufficiently interesting to warrant publication in Nature Communications.

Response: We thank the reviewer for their detailed comments and appreciation of our work. In the following, we address all comments point-by-point. In passing, we would like to note that we certainly work on THz-NSOT-driven Neel-vector switching of Mn₂Au, but this task is rather nontrivial and significantly beyond the scope of the current manuscript. Please see our response to comment C3.2 below.

There are, however, a few details that I thought required clarification.

C3.1: The initial alignment of the domains in the sample by an intense magnetic field pulse which is discussed in the main article and the methods. However, the dynamics are performed in a pump-probe setup, using a stroboscopic technique. Usually, the initial state is reset as the experiment is performed in a constant magnetic field. How do the authors ensure that the same initial state is achieved?

Response: For incident THz electric fields <250 kV/cm (corresponding to fields <15 kV/cm inside the sample), we find effects that scale linearly with the driving field. In other words, we work in the small-perturbation regime in which switching, which is a highly nonlinear process, does not occur.

Assume we start the experiment, and the first couple of pump pulses did switch the sample, the Neel vector would align perpendicular to the THz electric field, leading to a vanishing THz signal odd in L_0 . However, all data shown are the averaged results of 10^4 - 10^6 pump pulses interacting with the sample. Therefore, any irreversible effects of the very first pulses will not be visible.

To measure any irreversible effects, the sample needs to be reinitialized after each pump pulse, as mentioned by the referee and addressed in the next point.

Action: Please see action for C3.2

C3.2: In relation to the previous point, if the initial state cannot be reset then how could one determine whether switching has occurred? If, for example, the size of the electric field generated could be increased to the values predicted to result in switching (as shown in figure 4d by numerical modelling) then some domains would rotate at a certain delay and not be back switched for the next delay. The resulting stroboscopic values of Delta S might then be discontinuous when creating the time trace.

Response: We thank the reviewer for addressing this point. A true time-resolved observation of switching is nontrivial and requires two additional steps:

- 1) We need to further increase the THz field strength. This task is challenging as we already hit the limits of current table-top THz sources.
- 2) We need a procedure to temporally resolve an irreversible event (switching) with a repetitive pump-probe-type approach. To this end, we will use a modified sample system that allows us to reinitialize the sample after the action of each pump pulse. Our first choice is to work with Py-capped Mn₂Au films (Supplementary Note 5). This strategy, however, requires a more in-depth study both in terms of signal analysis and implications on the switching field strength in Mn₂Au (see also [9]), which is beyond the scope of the current work.

Action: We added text to “Results/Raw data” (line ~197), Supplementary Note 5 and the Conclusion (line 328).

C3.3: In the paragraph starting on line 272, is the scaling factor due to the difference between what the numerical model produces, something like the angle M makes as a function of time and Delta S?

Response: We thank the referee for this remark. In our model procedure, we calculate the Néel-vector deflection angle $\Delta\varphi_L$ of a single domain. The actually measured signal, however, contains the MLB coefficient and the average over domains:

$$\Delta S = a' L_0^2 \langle \sin(2\Delta\varphi_L) \rangle = a' L_0^2 (f_{0^\circ} - f_{180^\circ}) \sin(2\Delta\varphi_L) = C \sin(2\Delta\varphi_L)$$

where f_{0° and f_{180° is, respectively, the volume fraction of domains with the Néel vector parallel and antiparallel to \mathbf{u}_1 , with $f_{0^\circ} + f_{180^\circ} = 1$.

Note that $f_{0^\circ} - f_{180^\circ}$ is estimated to be of the order of 1-10% (see Supp. Note 11), but not precisely known. Likewise, the factor a' containing details of our detection setup and the magneto-optic coefficient is not known. All these unknowns are summarized by the factor $C := (f_{0^\circ} - f_{180^\circ}) a L_0^2$.

To summarize, our model provides $\Delta\varphi_L(t)$ of a single domain, but not the prefactor C , which contains lumped unknown sample information and is, thus, used as fit parameter. Note that all three curves in Fig. 4b are fitted jointly for one and the same C .

Action: We accordingly added text to the main text “Nonlinear regime and torkance” (line 293) and “Methods/Spatial averaging of the probe” (line 458).

References

1. Grishunin, K., et al., *Terahertz Magnon-Polaritons in TmFeO3*. ACS Photonics, 2018. **5**(4): p. 1375-1380.
2. Metzger, T.W.J., et al., *Effect of antiferromagnetic order on a propagating single-cycle THz pulse*. Applied Physics Letters, 2022. **121**: p. 252403.
3. Moriyasu, T., Wakabayashi, S., and Kohmoto, T., *Observation of Antiferromagnetic Magnons and Magnetostriction in Manganese Oxide Using Terahertz Time-Domain Spectroscopy*. Journal of Infrared, Millimeter, and Terahertz Waves, 2013. **34**(3-4): p. 277-288.
4. Nádvorník, L., et al., *Broadband Terahertz Probes of Anisotropic Magnetoresistance Disentangle Extrinsic and Intrinsic Contributions*. Physical Review X, 2021. **11**(2).
5. Gomonay, O., Jungwirth, T., and Sinova, J., *Narrow-band tunable terahertz detector in antiferromagnets via staggered-field and antidamping torques*. Physical Review B, 2018. **98**(10): p. 104430.
6. Reimers, S., et al., *Current-driven writing process in antiferromagnetic Mn2Au for memory applications*. Nature Communications, 2023. **14**(1).
7. Kim, Y., et al., *Investigation of THz birefringence measurement and calculation in Al2O3 and LiNbO3*. Applied Optics, 2011. **50**(18): p. 2906-2910.
8. Jourdan, M., et al., *Epitaxial Mn2Au thin films for antiferromagnetic spintronics*. Journal of Physics D: Applied Physics, 2015. **48**(38): p. 385001.
9. Hirst, J., et al. *Simulations of Magnetization Reversal in FM/AFM Bilayers With THz Frequency Pulses*. 2023. arXiv:2304.12969 DOI: 10.48550/arXiv.2304.12969.

Reviewers' Comments:

Reviewer #1:

Remarks to the Author:

The authors provided detailed response to the comments by Reviewers and made necessary modifications to the main text and supplementary information.

In my opinion, the manuscript can be accepted to Nature Communication in the present form.

Reviewer #2:

Remarks to the Author:

The authors have very satisfactorily answered all the comments from all the reviewers. I have reviewed each of them and highly recommend publishing this work.

Reviewer #3:

Remarks to the Author:

The authors have addressed all of my comments and I would recommend the work for publication as is.

REVIEWERS' COMMENTS

Reviewer #1 (Remarks to the Author):

The authors provided detailed response to the comments by Reviewers and made necessary modifications to the main text and supplementary information.

In my opinion, the manuscript can be accepted to Nature Communication in the present form.

Reviewer #2 (Remarks to the Author):

The authors have very satisfactorily answered all the comments from all the reviewers. I have reviewed each of them and highly recommend publishing this work.

Reviewer #3 (Remarks to the Author):

The authors have addressed all of my comments and I would recommend the work for publication as is.

We would like to thank all reviewers for their appreciation of our work, as well as for their questions and comments, which allowed us to significantly improve our manuscript.